# THE PLUG AND PLAY OF LANGUAGE MODELS FOR TEXT-TO-IMAGE GENERATION

## ABSTRACT

Text-to-image (T2I) models enable controllable image generation through user-provided captions. A text encoder is typically used to map captions to a latent space, and it has been shown to be critical for model's performance. However, replacing or upgrading the text encoder in a T2I model is challenging due to the tight bond between the current encoder and the image decoder. It requires training the model from scratch, which can be prohibitively expensive. To address this problem, we introduce a more efficient approach to align a pre-trained language model with the latent space of an existing T2I model. We propose a Model Translation Network (MTN) and a new training objective to align the representation spaces of the two text encoders using only a corpus of unlabeled text. We empirically find that MTN can be trained efficiently and can boost the performance of existing T2I models by upgrading their text encoder. Moreover, we find that MTN can align multilingual language models such as XLM-Roberta, thus allowing existing T2I models to generate high-quality images from captions beyond English.

## 1 INTRODUCTION

Text-to-image (T2I) generative models have made great progress in the last few years thanks to algorithmic advances and the availability of large-scale paired training datasets (Ramesh et al., 2022; Yu et al., 2022a; Saharia et al., 2022; Rombach et al., 2022). Diffusion-based T2I generative models in particular have achieved remarkable results in terms of image quality (Ho & Salimans, 2022; Nichol et al., 2021). Despite these strong results, controllable generation for these methods is still challenging: generated images are often not faithful to the captions, compositional capabilities are lacking, and prompt engineering is often required to achieve the desired results (Parsons, 2022). Moreover, most large-scale models have only been trained on English captions, greatly limiting their use across the world.

To improve T2I models, recent research suggests that the ability of their text encoders to understand and represent text is critical and is a bottleneck for their image generation performance (Saharia et al., 2022; Croitoru et al., 2022). Text encoders in existing T2I models are often trained only on short image captions, and their performance on complex prompts can be largely constrained by the quality of the features extracted by the text encoder (Rombach et al., 2022). However, upgrading the text encoder for an existing T2I model is challenging because the representation spaces of the text encoder and image generator are tightly coupled (Rombach et al., 2022; Ramesh et al., 2022). Training only the text encoder on a more complex and representative text corpus would break this alignment, hindering the final image generation performance. Training the entire T2I model from scratch (perhaps with higher quality image-caption pairs) would be prohibitively expensive (Edwards, 2022).[1]

To solve this problem, we propose a method that can efficiently align off-the-shelf pre-trained language models with image encoders of existing diffusion-based T2I models. With this method, the existing text encoder of T2I models can be replaced with a more powerful language model or even one for a language other than English as illustrated in Fig. 1. Crucially, the representation alignment between text and image encoders is maintained without retraining from scratch. We refer to our proposed method as Model Translation Network (MTN). MTN follows an encoder-decoder structure. Specifically, given a trained T2I model and a pre-trained language model that we want to replace, the

---

[1]The cost of training a Stable Diffusion model is around 600K USD.

Figure 1: Illustration of our desired modularized T2I generation. With the proposed Model Translation Network (MTN), the existing image generators can be bridged to off-the-shelf language models to expand their functionalities, *i.e*, multilingual generation, within a limited budget.

encoder of MNT firstly aligns the representation space of the pre-trained language model with that of the T2I model's image generator, by minimizing both element-wise and global-wise discrepancy. Then, the decoder of MNT takes the aligned text representations as inputs, and maps them back to the original representation space of the pre-trained language model by minimizing the reconstruction loss. The reason why we need this decoder during training is that some recent research reveals that training to align existing models inevitably decreases feature discriminability (Chen et al., 2019; Cui et al., 2020). Therefore we need to preserve rich semantics captured by the pre-trained language model during the alignment training, ensuring that a decoder can recover the original representation space of the pre-trained language model from the aligned one. The entire training of MTN requires only a corpus of unlabeled text. At inference time, only the encoder of MTN is applied on top of the pre-trained language model for representation alignment.

To verify the effectiveness of the proposed framework, we have applied a stronger language model, *i.e*, T5-3B (Raffel et al., 2020), to upgrade existing text encoders of the Latent Diffusion Model (Rombach et al., 2022). The competitive improvements in the FID score and user study ranking reveal the benefits of our model over the baselines. Furthermore, our model has the potential to bring new functionalities into existing T2I models, such as multilingual text-to-image generation. We empirically find that MTN can align a multilingual language model such as XLM-Roberta-L (Conneau et al., 2019) with existing T2I models. It enables the existing image generator to understand text beyond English, such as French and Chinese, and to generate high-quality images accordingly.

Our contributions can be summarized as follows:

- To the best of our knowledge, this is the first work to consider the problem of efficiently aligning a pre-trained language model with a pre-trained T2I diffusion model.

- Extensive experiments on text-to-image generation benchmarks demonstrate the superiority of our model over the baseline LDM method on both image quality and language controllability.

- Our framework also enables text-to-image generation beyond English prompts without the need of multilingual image-text pairs for retraining.

## 2   RELATED WORKS

### 2.1   TEXT-TO-IMAGE GENERATION

Generative Adversarial Nets (GANs) (Goodfellow et al., 2014) is one of the major frameworks in text-to-image generation. The method in (Reed et al., 2016) was an early stage approach to use a GAN to train a text-to-image generation network. Since then, other GAN based methods, *e.g*, Stack-GAN (Zhang et al., 2017), Attn-GAN (Xu et al., 2018) and SD-GAN (Yin et al., 2019), have obtained promising results. In addition, recent works showed more improvements on the generation quality. DM-GAN (Zhu et al., 2019) improved text-to-image performance by introducing a dynamic memory component. DF-GAN (Tao et al., 2022) designed a fusion module to fuse text and image features. LAFITE (Zhou et al., 2021) took advantage of CLIP's model to construct pseudo image-text pairs, and proposed a GAN model to learn text-image pairs.

Auto-regressive transformer is another major framework in text-to-image generation. DALL-E (Ramesh et al., 2021) and CogView (Ding et al., 2021) adopted an autoregressive transformer (Vaswani et al., 2017) to train the correspondences between visual tokens and text tokens. Parti (Yu et al., 2022a) used a powerful visual encoder, ViT-VQGAN (Yu et al., 2022b), to improve results upon DALL-E. The method in (Gafni et al., 2022) is similar to CogView and DALL-E, while introducing additional controlling elements to improve the tokenization process. NÜWA (Wu et al., 2022b) and NUWA-infinity (Wu et al., 2022a) trained autoregressive visual synthesis model to support both text-to-image and text-to-video generation tasks.

Concurrently, diffusion models (Sohl-Dickstein et al., 2015; Song & Ermon, 2019; Song et al., 2020) became a main research focus. In such methods, noise is added to an image, and a score network is trained to denoise and recover the input image. GLIDE (Nichol et al., 2021) performed guided inference with and without a classifier network to generate high-fidelity images. DALL-E 2 (Ramesh et al., 2022) and Imagen (Saharia et al., 2022) set new state-of-the-art results in the text-to-image generation. In DALL-E 2, a prior to produce CLIP image embeddings conditioned on text was learned, and a diffusion model was used to decode the image embeddings to an image. In Imagen, a frozen large-scale text encoder T5-XXL (Raffel et al., 2020) was adopted to generate embeddings, and a cascade of conditional diffusion models was used to map these embeddings to images of increasing resolutions. Latent diffusion model and stable diffusion model (Rombach et al., 2022) are state-of-the-art methods that apply the diffusion model on the latent embedding space as in (Sinha et al., 2021; Vahdat et al., 2021). These methods are computationally friendly while achieving impressive results.

## 2.2 KNOWLEDGE DISTILLATION AND CROSS-LINGUAL ALIGNMENT

Knowledge distillation is a technique to guide the training of a student network with a reference teacher network (Buciluǎ et al., 2006). (Hinton et al., 2014) applied probabilities of the teacher as a target for the student to learn. Since then, many methods around the idea of "what knowledge to be distilled" have been proposed. Popular definitions include distance of features (Romero et al., 2015), map attention (Zagoruyko & Komodakis, 2017), distribution-wise distillation (Passalis & Tefas, 2018) and inter-sample distance structure (Tung & Mori, 2019), etc. Despite the great success, they mainly focus on the classification task with discrete outputs and pay less attention to representation protection. Translating dense and high-dimensional features between large-scale language models requires new solutions.

Cross-lingual alignment methods attempt to align the representation across different languages based on a multilingual text model (Kulshreshtha et al.; Cao et al., 2020). (Kulshreshtha et al.) has comprehensively explored the different alignment methods, including rotation and finetuning alignment. (Cao et al., 2020) has modeled the contextual token-wise embeddings to aggregate richer information for alignment. Our paper considers another problem in aligning different text models with the same or parallel data. This is essentially different from cross-lingual alignment, with more complicated and challenging mismatches such as feature dimensions and variant token lengths.

## 3 PRELIMINARY

### 3.1 LATENT DIFFUSION MODEL

The Latent Diffusion Model (LDM) (Rombach et al., 2022) and its extension, *i.e*, Stable Diffusion, which are variants of Denoising Diffusion Probabilistic Model (DDPM) (Ho et al., 2020) family, are selected as our baseline models due to their excellent balance in efficiency and visual quality.

In general, LDM is composed of two stages. Firstly, there is an auto-encoder (AE) pre-trained by enormous images with the regularization in either KL-divergence or vector quantization (Van Den Oord et al., 2017; Agustsson et al., 2017). An encoder network, *i.e*, $E$, is applied for latent feature extraction as $z = E(x)$, which can be mapped back to image space by a decoder network. The input images $x$ and reconstructed images $\hat{x}$ are almost identical $x \approx \hat{x}$.

Secondly, LDM trains a diffusion model in the latent space (Sinha et al., 2021; Vahdat et al., 2021). It follows the standard DDPM (Ho et al., 2020) with denoising loss and uses U-net (Ronneberger et al., 2015) as the image decoder as in (Song & Ermon, 2019). To enable generative controllability,

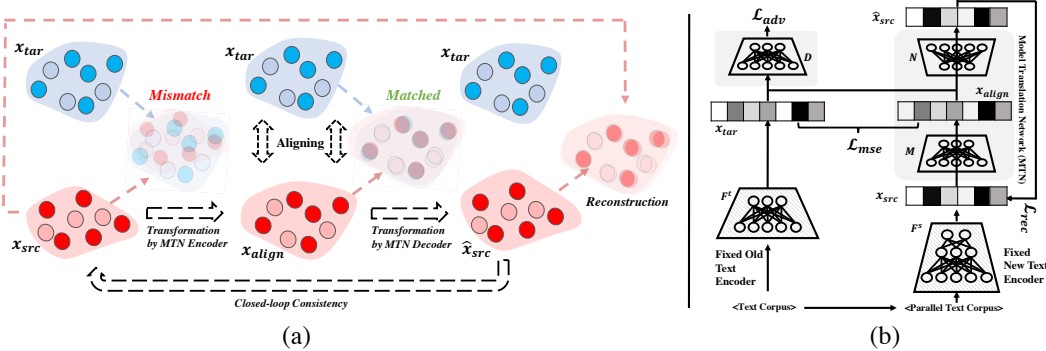

Figure 2: (a): illustration of features transformation throughout the model translation. (b): the general pipeline and learning objectives of our proposed model translation network (MTN).

LDM has applied multiple conditioning signals ($y$) such as text, mask, or layout, encoded aside and injected into the U-net, with the help of cross-attention layers. This can be formulated as:

$$\mathcal{L}_{LDM} := \mathbb{E}_{z \sim E(x), \varepsilon \sim N(0,1), t, y} \left[ \|\varepsilon - \varepsilon_\theta(z_t, t, c_\phi(y))\| \right],  \tag{1}$$

where $t$ represents the time step, $z_t$ is the noise corrupted latent tensor at time step $t$, and $z_0 = E(x)$. $\varepsilon$ is the unscaled Gaussian noise, $c_\phi$ is the conditioning network parameterized by $\phi$ and $\varepsilon_\theta$ is the Unet-like denoising network (*a.k.a*, image decoder). The parameters of both conditioning and denoising networks, *i.e*, $\theta, \phi$, are trained by the LDM loss. During inference, clean images can be generated via classifier-free guidance (Ho & Salimans, 2022) as:

$$\hat{\varepsilon}_\theta(z_t|y) = \varepsilon_\theta(z_t) + s \cdot (\varepsilon_\theta(z_t, c_\phi(y)) - \varepsilon_\theta(z_t)),  \tag{2}$$

where $s$ is the guidance weight to balance text controllability and image fidelity.

The basic T2I LDM model is trained on Laion-400M (Schuhmann et al., 2021) dataset. The Stable Diffusion Model (SDM) was trained with more epochs and training data, *i.e*, Laion2B-en and Laion-high-resolution.

## 3.2 LANGUAGE MODELS FOR T2I

LDM uses Bert (Devlin et al., 2018) as the text encoder jointly trained with the image decoder using the DDPM loss. SDM uses a pre-trained CLIP (Radford et al., 2021) text encoder frozen during training. It has been found that increasing the size of the text encoder is very helpful for performance (Saharia et al., 2022). In order to upgrade existing models, our goal is to be able to efficienlty plug in more advanced language models such as T5-3B (Raffel et al., 2020) with more parameters or XLM-Roberta (Devlin et al., 2018) for multilingual T2I.

## 4 MODEL TRANSLATION NETWORK (MTN)

The text encoder is one of the key components of T2I models, as generation requires precise and fine-grained text embedding for guidance. The overall performance can be greatly boosted by increasing the size of the text encoder (Saharia et al., 2022), however, upgrading an existing text model to a more powerful one is challenging. Because existing T2I models are not modular, directly replacing (or augmenting with) another pre-trained language model does not work as seen in Fig. 9 (a). The key technical challenge is the mis-alignment between the new language model and the old image decoder. Moreover, joint finetuning also falls short due to catastrophic forgetting occuring when updating well trained parameters.

Considering these two different text models as source and target domains, we apply a neural network to learn to align. As shown in Fig. 2, to efficiently upgrade the text encoder and simultaneously protect existing knowledge, this paper has presented a general framework called Model Translation Network (MTN). MTN works as an additional module to bridge the new language model and the old image decoder as seen in Fig. 1.

## 4.1 OBJECTIVES

Given a set of text corpus $\mathcal{X} = \{x_{ij}|i = 1, ..., M, j = 1, ..., N\}$ with $M$ as length of tokens and $N$ as total quantity, the source language model $F^s$ and a target text encoder $F^t$ have mapped the raw text data into embeddings denoted as $\mathcal{S} = \{s_{ij}|i = 1, ..., M_s, j = 1, ..., N\}$ where $s_{ij} = F^s(x_{ij}) \in \mathbb{R}^{d_s}$ and $\mathcal{T} = \{t_{ij}|i = 1, .., M_t, j = 1, ..., N\}$ where $t_{ij} = F^t(x_{ij}) \in \mathbb{R}^{d_t}$. Due to the different models, there is severe distribution mismatch between the source and target features $p(s) \neq p(t)$. The MTN is an autoencoder whose encoder $M(\cdot, \Theta_M)$ learns to map the new text features from a source language model, *i.e*, $s \in \mathcal{S}$, to align with the current target text encoder, *i.e*, $t \in \mathcal{T}$, where $p(M(s, \Theta_M)) \approx p(t)$. Therefore, this translation (*i.e*, $F^s \xrightarrow{\Theta_M} F^t$ ) enables the image generator to understand the new text features coming from the new language model without finetuning. To achieve this aim, we consider it as a domain adaptation (Ganin et al., 2016; Qin et al., 2019; Long et al., 2018) problem and apply both element-wise and distribution-wise alignment losses, which includes the minimization of the mean square error (MSE) loss, *i.e*, $\mathcal{L}_{mse}$, and the adversarial loss, *i.e*, $\mathcal{L}_{adv}$, measured over a discriminator network, *i.e*, $D(\cdot, \Theta_D)$:

$$\mathcal{L}_{mse}(\Theta_M) := \mathbb{E}_{x_t \sim X_t, x_s \sim X_s}[||F^t(x_t) - M(F^s(x_s), \Theta_M)||_2^2], \tag{3}$$

$$\mathcal{L}_{adv}(\Theta_D, \Theta_M) := \mathbb{E}_{x_t \sim X_t}[\log D(F^t(x_t), \Theta_D)] + \mathbb{E}_{x_s \sim X_s}[\log 1 - D(M(F^s(x_s), \Theta_M), \Theta_D)], \tag{4}$$

where $X_s$ and $X_t$ denote the source and target inputs respectively and they can be the same prompts in English (*e.g*, LDM-Bert $\rightarrow$ T5-3B (Raffel et al., 2020)) or in bilingual but parallel content for multilingual T2I generation.

Some insightful investigations in transfer learning reveal that the cross-domain alignment will inevitably decrease feature discrimination (Chen et al., 2019; Cui et al., 2020). To project the rich semantics necessary for model upgrading, we further apply a decoder network, *i.e*, $N(\cdot, \Theta_N)$, for feature reconstruction:

$$\mathcal{L}_{rec}(\Theta_M, \Theta_N) := \mathbb{E}_{x_s \sim X_s}[||x_s - N(M(F^s(x_s), \Theta_M), \Theta_N)||_2^2]. \tag{5}$$

MTN is trained in an end-to-end manner and the details of which can be referred to in **Algorithm 1** and Sec. 5. (Ma et al., 2022) introduced that a universal learning engine should seek a compressive closed-loop transcription with good property in structure-preserving. Our cross-model mapping also requires a similar functionality. MTN desires stability after a loop whose input and output should keep almost equivalent, *i.e*, $x \approx \hat{x} \approx \hat{\hat{x}} \approx ...$, due to the constraint of reconstruction loss. Therefore, inputting the reconstructed data $\hat{x}$ to MTN is expected to be consistent with the previous loop. This is helpful for accelerating MTN training.

## 4.2 ARCHITECTURE OF MODEL TRANSLATION NETWORK

The input data to alignment can be represented as the sequence of tokens: $\mathrm{X}_{i,*} = \{x_{i,1}, x_{i,2}, ..., x_{i,S}\}$ where $\mathrm{X}_{i,*} \in \mathcal{X}$ and $x_{i,j} \in \mathbb{R}^C$, with sequence length $S$ and token dim $C$. Inspired by the methods of image super-resolution (Lim et al., 2017; Ahn et al., 2018) to densely regress target data under different input-output dimensions, the translator network stacks three sub-nets in sequence, as seen in Fig. 3. The head net is only used for simple feature transformation. The body net has many residual modules for fine-grained representation learning. The tail network

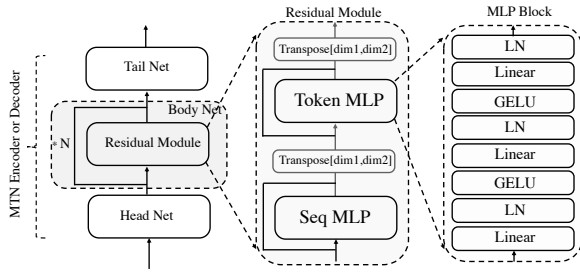

Figure 3: Detailed architecture of Model Translation Network.

is appended and is mainly employed for dimension conversion to match the target tensors. Every residual module is implemented based on the MLP-based token mixer (Tolstikhin et al., 2021; Ma et al., 2021) with high efficiency and strong ability in representation learning. Within the MLP-mixer block, it consists of a Token Mixer and a Sequence Mixer for orthometric purposes. The

Token Mixer learns the representation of each token with shared MLP, and the Sequence Mixer is designed for learning the same channel in sequential tokens:

$$\text{TokMixer}: \quad \text{U}_{\cdot,i} = \text{X}_{\cdot,i} + \text{W}_2\sigma(\text{W}_1 * \text{LN}_{\gamma_1,\beta_1}(\text{X})_{\cdot,i}), i = 1, ..., C \tag{6}$$

$$\text{SeqMixer}: \quad \text{X}_{j,\cdot} = \text{X}_{j,\cdot} + \text{W}_4\sigma(\text{W}_3 * \text{LN}_{\gamma_2,\beta_2}(\text{U})_{j,\cdot}), j = 1, ..., S \tag{7}$$

where $\text{X}_{\cdot,\cdot}$ is the feature, $\text{LN}_{\gamma,\beta}$ denotes layer normalization parameterized by $\gamma$ and $\beta$, and $\sigma$ indicates the non-linear operator such as GELU (Hendrycks & Gimpel, 2016).

## 5 EXPERIMENTS

In this section, we comprehensively evaluate our proposed framework for the monolingual and multilingual text-to-image generation upon upgrading the text encoder of pre-trained models. More details can be referred to in **Appendix**.

### 5.1 EXPERIMENTS SETUP

Upon the shoulders of giants, this paper applies the Latent-Diffusion Model (LDM) and Stable Diffusion Model (SDM) (Rombach et al., 2022) as the baselines with a similar model but different text encoders and scalabilities. LDM uses a Bert (Devlin et al., 2018) as the text encoder that is jointly updated with the image decoder based on DDPM loss, while SDM uses a frozen pre-trained CLIP text encoder during training. The LDM is designed for the images in $256 \times 256$ and SDM targets $512 \times 512$ images. Considering our limitations in data and computation resources, the experiments on monolingual T2I are based on LDM with a lower cost of text encoder upgrading than that of SDM. We implement the multilingual T2I generation upon the SDM.

**Implementation.** The MTN is implemented as the Fig. 3 with an encoder $M$ and a decoder $N$ in symmetrical architecture consisting ~51M parameters. For simplicity in denotation, we refer MTN encoder as MTN in the following experiments, whose body net consists of 5 residual modules as default. We take the AdamW (Loshchilov & Hutter, 2017) as the optimizer based on PyTorch (Paszke et al., 2019). The learning rate is assigned as $1 \times 10^{-4}$ for MTN training and $1 \times 10^{-6}$ for LDM-T5-MTN finetuning. The MTN model is trained on Nvidia-A100-40GB, requiring 1 to 5 GPU days.

**Language Models.** We apply the T5-3B (Raffel et al., 2020) to upgrade the text encoder of LDM Bert. T5-3B has 6x more parameters than LDM Bert and is trained by an enormous text corpus beyond the image captions. Furthermore, to enable multilingual T2I generation, we choose the XLM-Roberta-L (Devlin et al., 2018) as the new text encoder to let SDM image decoder understand languages beyond English.

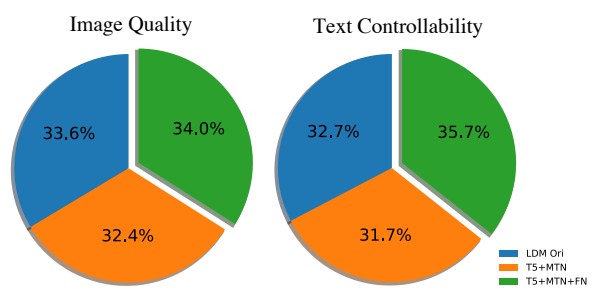

Figure 4: User study of text controllability and image quality with the prompts from Drawbench (Saharia et al., 2022) and Winoground (Thrush et al., 2022).

**Datasets.** We have collected the text data for MTN training. In monolingual T2I, only English texts are required, and we extract 18M captions from Laion-400M as training corpus, and more data will be more helpful. Multilingual MTNs require parallel texts in different languages. These data are collected by Wikimarix (Schwenk et al., 2019) and Googletrans with ~2M sentences for each.

**Ablations. T5+MTN** refers to the LDM model with the stacking of T5 and MTN as the text encoder where the LDM image decoder is unchanged. **T5+FT** represents finetuning of the LDM image decoder with DDPM loss based on text conditions from a fixed T5. **T5+MTN+FT** is the final version, including both MTN and joint finetuning with the image decoder. The finetuning takes ~ 100 GPU days of 116M image-text pairs filtered from Laion-400M by BLIP score (Li et al., 2022).

### 5.2 MONOLINGUAL TEXT-TO-IMAGE GENERATION

Monolingual T2I generation focuses on the comparison with LDM given the English text prompts. We have decomposed our pipeline into two stages: alignment and finetuning. MTN helps to align

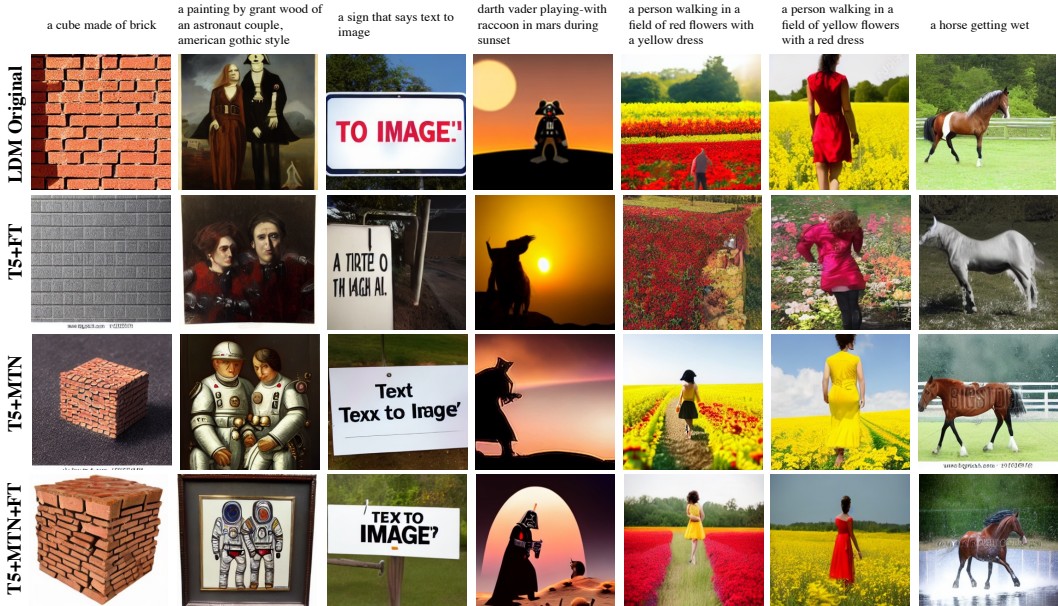

Figure 5: Monolingual generation in resolution 256 × 256 with guidance weight 7.5 and DDIM steps 200. The details of each method can be referred to in Sec. 5.1.

the new language model whose potential can be better exploited by further finetuning of image text pairs. This section verifies the benefits of our vanilla and finetuned models in both quantitative and qualitative evaluations.

### 5.2.1 QUANTITATIVE EVALUATION

Tab. 1 reported the zero-shot FID score (Heusel et al., 2017) on COCO dataset (Lin et al., 2014) based on the pyorch-fid (Seitzer, 2020) package. Our re-implementation score of LDM is slightly higher than the paper's one (Rombach et al., 2022), which may because of different implementations. The table shows the minor drop of T5+MTN from LDM since our current MTN model still leaves some incorrect alignments, which could be fixed by either joint finetuning or collecting more text corpus to train MTN. With the ∼100 GPU-days finetuning of a subset of the Laion dataset (*i.e*, 116M image-text pairs), the overall performance has been considerably improved, even outperforming the LDM by certain margins.

We have also conducted a user study to compare the image quality and text controllability in Fig. 4 by Amazon Turk (Buhrmester et al., 2011). There are ∼1K prompts collected from Drawbench (Saharia et al., 2022) and Winoground (Thrush et al., 2022), which comprehensively cover various scenarios and numerous difficult cases. We report the percentages of users' votes for the three methods in Fig. 4. The study of image quality is designed to evaluate the resolution, cleanness, and smoothness of object boundaries, where our method shows fragile superiority. In the user study of controllability, ours outperformed the LDM by 3%, which is a non-trivial improvement. Moreover, the p-value [2] is computed as 0.04 with three repeats of user study that is statistically significant according to the criteria of <0.05.

Table 1: FID on COCO. * indicates our re-implemented results. ZS denotes zero-shot.

| Method | FID↓ | ZS |
|---|---|---|
| CogView (Ding et al., 2021) | 27.10 | ✗ |
| LAFTTE (Zhou et al., 2021) | 26.94 | ✗ |
| GLIDE (Nichol et al., 2021) | 12.24 | ✗ |
| Make-A-Secne (Gafni et al., 2022) | **11.84** | ✗ |
| LDM (Rombach et al., 2022) | 12.63 | ✓ |
| LDM* | 13.55 | ✓ |
| T5+FT | 23.30 | ✓ |
| T5+MTN | 14.32 | ✓ |
| T5+MTN+FT | 12.05 | ✓ |

[2]https://docs.scipy.org/doc/scipy/reference/generated/scipy.stats.binomtest.html

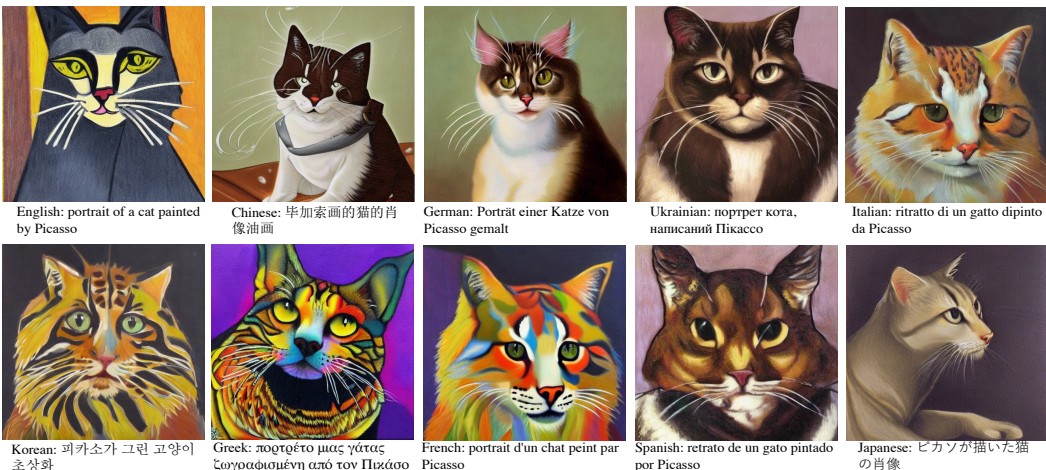

English: portrait of a cat painted by Picasso

Chinese: 毕加索画的猫的肖像油画

German: Porträt einer Katze von Picasso gemalt

Ukrainian: портрет кота, написаний Пікассо

Italian: ritratto di un gatto dipinto da Picasso

Korean: 피카소가 그린 고양이 초상화

Greek: πορτρέτο μιας γάτας ζωγραφισμένη από τον Πικάσο

French: portrait d'un chat peint par Picasso

Spanish: retrato de un gato pintado por Picasso

Japanese: ピカソが描いた猫の肖像

Figure 6: Multilingual generation results in resolution $512 \times 512$ of XLM-Roberta + MTN + SDM decoder (sd-v1-4) with the same caption, *i.e*, "portrait of a cat painted by Picasso". With the help of different MTNs and multilingual text encoder, the SDM decoder can support different languages including Japanese, Ukraine, German, Italian, Korean, Greek, Chinese, French and Spanish. The guidance weight is assigned as 7.5 and PLMS (Liu et al., 2022) sampling steps are 50.

### 5.2.2 VISUAL COMPARISON

In Fig. 5, we have exhibited some generated images of four different methods. By comparing T5+FT and T5+MNT+FT with similar finetuning costs (100 GPU-days), we can conclude that the proposed MNT helps save much effort in plugging a new language model due to the high-quality images. Moreover, the T5+MNT+FT shows stronger controllability in many cases compared with LDM, such as "a cube made of brick" and "a horse getting wet", whereas the LDM's results fail to display the concepts of "cube" and "wet".

### 5.3 MULTILINGUAL TEXT-TO-IMAGE GENERATION

Our proposed framework is very general and generic to many language models, including the multilingual backbones. We choose the recent SDM as our base model with the pre-trained clip-vit-large-patch14 as the text encoder to be upgraded by the multilingual one, *i.e*, XLM-Roberta-L. The MTN helps to align such two backbones with the parallel corpus as training data. Thus, the old SDM image decoder can successfully understand the text embeddings of other languages encoded by MTN and XLM-Roberta-L without any additional parameters updating except MTN.

### 5.3.1 VISUAL COMPARISON

Fig. 6 shows the example of multilingual generation results with the prompt "portrait of a cat painted by Picasso" in many languages. Most of the results reveal the content of "portrait of a cat painted". The "Picasso" is a challenging notion that rarely appears in our training data. Thus, more high-quality parallel data will be greatly helpful. Compared with those finetuned over multilingual image text pairs, our solution is far more efficient and cheaper.

### 5.3.2 HYBRID MULTILINGUAL GENERATION

The hybrid multilingual generation is more challenging that both the original SDM and third-party translation toolbox cannot address. Our proposed MTN has surprising capability in this problem. According to the results in Fig. 7, the multilingual backbone bonded with MTN gives the precise guide to the SDM decoder even with the prompts in three different languages and randomly mixed.

### 5.4 ALIGNMENT ANALYSIS

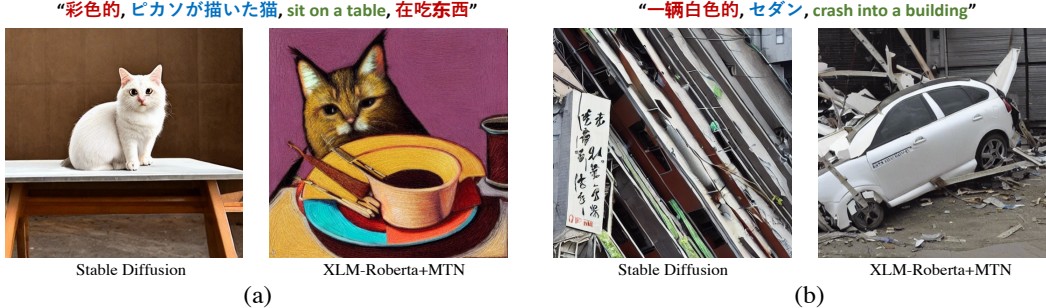

Figure 7: Hybrid multilingual generation in resolution of 512 × 512. There are three-different-language texts in the input caption including Chinese, Japanese and English. The caption of (a) is "colorful, a cat painted by Picasso, sit on a table, is eating food" and the caption of (b) is "a white, sedan, crash into a building". With our MTN infused ahead, the XLM-Roberta can guide SDM decoder to generate reasonable results where the original SDM fails to work.

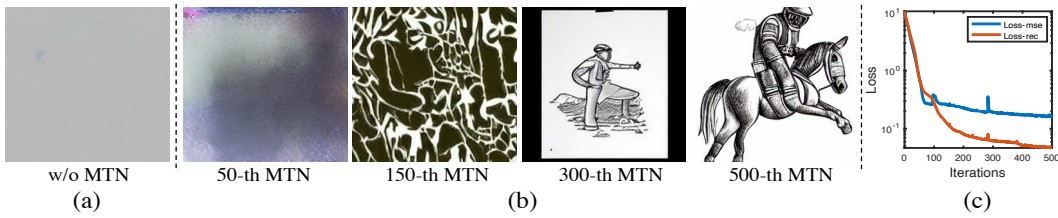

Figure 9: Alignment analysis. (a): Initial result of direct replacement of a new text encoder, *i.e*, LDM-Bert → T5-3B. (b): Results of MTN in different iterations with the new text backbone. The prompt is "an astronaut riding a horse as a pencil drawing". (c): Curve of $\mathcal{L}_{mse}$ and $\mathcal{L}_{rec}$ in training.

To understand how MTN maps the cross-model features, we have visualized its trajectory with the generative results in different iterations in Fig. 9. In the beginning, the old image decoder could not understand the new text embeddings at all. Then, certain patterns such as "pencil drawing" appear after 300 iterations of training with a significant decrease in losses. Such a gap can finally be sealed at the end where both $\mathcal{L}_{mse}$ and $\mathcal{L}_{rec}$ converged to small values.

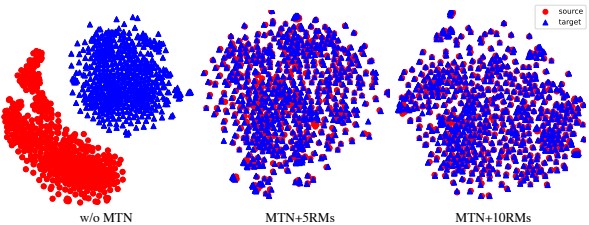

Figure 8: t-SNE (Van der Maaten & Hinton, 2008) cross-model feature map visualization.

### 5.5 T-SNE FEATURE MAPS WITH DIFFERENT MTN SIZES

Fig. 8 illustrates the feature maps of different text encoders. Source features are collected from LDM-Bert and target ones are achieved by T5-3B with the shared prompts from Drawbench and Winoground. RMs represents the number of the residual modules. We can see that MTNs' features are precisely aligned with those of the source text model. There is no much difference between larger and smaller MTNs. Considering efficiency, we choose the MTN-5RMs as the default model.

## 6 CONCLUSION

Infusing the pre-trained language models into the existing T2I image generator is an existing direction toward a more powerful AI system. However, the current text encoder cannot be easily upgraded due to the tight match. This paper attempts to break such a strong constraint of the corresponding image-text models towards flexible modularization and efficient upgrading. To address the severe misalignment, we have proposed the model translation network (MTN) with both objectives for cross-model alignment and originality protection. Empirically, it is beneficial for the overall performance and enables versatile functionalities such as multilingual generation within a limited budget. We wish this work to be inspirable to the community in large-scale AI system design.

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

## ETHICS STATEMENT

This paper strictly follows the ICLR ethical research standards and laws. All datasets we employed are publicly available, and all related publications and source codes are cited appropriately.

## REPRODUCIBILITY STATEMENT

We adhere to ICLR reproducibility standards and ensure the reproducibility of our work in multiple ways, including:

- Besides the codes, we will share the our processed data and model checkpoints to the public.
- We clearly present the design of MTN and joint finetuning in Section 5.1 and Section 4.2
- Detailed framework and more experiments are presented in Supplementary Materials.

# A  DETAILS OF MODEL TRANSLATION NETWORK (MTN)

Model Translation Network (MTN) is the key contribution of this paper, which needs to address both cross-model alignment and feature protection. In this section, we will provide more details of the MTN, including implementation/configuration, analysis of sizes, and other ablation studies.

## A.1  ARCHITECTURE AND CONFIGURATION

We have introduced the basic architecture of MTN in Sec. 4.2 in the main paper. MTN could have variants in different sizes whose configurations can be referred to in Tab. 2. The MTN with three residual modules is the smaller one, with 34M parameters in its encoder. The MTN-5RMs has 51M parameters. In general, larger-size models inevitably result in slower speed and more computation costs. Considering the sizes of the image decoder and text encoder as the basis, the MTN, working as an injected module, should not be larger than these major characters. Otherwise, the inference speed will be significantly slower, and it may need huge computations in the finetuning stage. We empirically found that assigning RM as five is a satisfactory choice. The MTN-3RMs seems too weak in representation learning. Except for the MLP-mixer, the self-attention-based model may also be applied for implementing MTN. Exploring more suitable architectures of MTN is our future work.

Table 2: Model configurations for our MTN. We introduce two configurations of MTN-3RMs and MTN-5RMs.

| Stage | Dimensions | Block | MTN-3RMs | | MTN-5RMs | |
|---|---|---|---|---|---|---|
| Head Net | 77 | Token MLP | $\begin{bmatrix} \text{Linear}, \text{LN}, \sigma \\ \text{Linear}, \text{LN}, \sigma \\ \text{Linear}, \text{LN} \end{bmatrix}$ | | $\begin{bmatrix} \text{Linear}, \text{LN}, \sigma \\ \text{Linear}, \text{LN}, \sigma \\ \text{Linear}, \text{LN} \end{bmatrix}$ | |
| | 1024 | Sequence MLP | $\begin{bmatrix} \text{Linear}, \text{LN}, \sigma \\ \text{Linear}, \text{LN}, \sigma \\ \text{Linear}, \text{LN} \end{bmatrix}$ | | $\begin{bmatrix} \text{Linear}, \text{LN}, \sigma \\ \text{Linear}, \text{LN}, \sigma \\ \text{Linear}, \text{LN} \end{bmatrix}$ | |
| Body Net | 77 | Token MLP | $\begin{bmatrix} \text{Linear}, \text{LN}, \sigma \\ \text{Linear}, \text{LN}, \sigma \\ \text{Linear}, \text{LN} \end{bmatrix}$ | $\times\,3$ | $\begin{bmatrix} \text{Linear}, \text{LN}, \sigma \\ \text{Linear}, \text{LN}, \sigma \\ \text{Linear}, \text{LN} \end{bmatrix}$ | $\times\,5$ |
| | 1024 | Sequence MLP | $\begin{bmatrix} \text{Linear}, \text{LN}, \sigma \\ \text{Linear}, \text{LN}, \sigma \\ \text{Linear}, \text{LN} \end{bmatrix}$ | $\times\,3$ | $\begin{bmatrix} \text{Linear}, \text{LN}, \sigma \\ \text{Linear}, \text{LN}, \sigma \\ \text{Linear}, \text{LN} \end{bmatrix}$ | $\times\,5$ |
| Tail Net | 77 | Token MLP | $\begin{bmatrix} \text{Linear}, \text{LN}, \sigma \\ \text{Linear}, \text{LN}, \sigma \\ \text{Linear}, \text{LN} \end{bmatrix}$ | | $\begin{bmatrix} \text{Linear}, \text{LN}, \sigma \\ \text{Linear}, \text{LN}, \sigma \\ \text{Linear}, \text{LN} \end{bmatrix}$ | |
| | 1280 | Sequence MLP | $\begin{bmatrix} \text{Linear}, \text{LN}, \sigma \\ \text{Linear}, \text{LN}, \sigma \\ \text{Linear}, \text{LN} \end{bmatrix}$ | | $\begin{bmatrix} \text{Linear}, \text{LN}, \sigma \\ \text{Linear}, \text{LN}, \sigma \\ \text{Linear}, \text{LN} \end{bmatrix}$ | |

## A.2  EFFECTS OF MTN SIZES

Fig. 10 shows a visual comparison of some example prompts over three different methods. The LDM Ori refers to the original LDM model with the checkpoint shared in its repo. MTN-3RMs and MTN-5RMs are our models with 3 or 5 residual modules inside their body nets. The text encoder is replaced as the T5-3B, and the image decoder is the same as LDM Ori. Both MTN models are trained by the same text corpus, including 18M sentences. From the figure, we can notice that MTN-3RMs fails on some complicated prompts such as "a virus monster is playing guitar, oil on canvas" and "there is a penguin with a dog head standing" where the MTN-5RMs performs well. Therefore, we can conclude that deep MTN is necessary for precise alignment.

|  | **LDM Ori** | **MTN-3RMs** | **MTN-5RMs** |

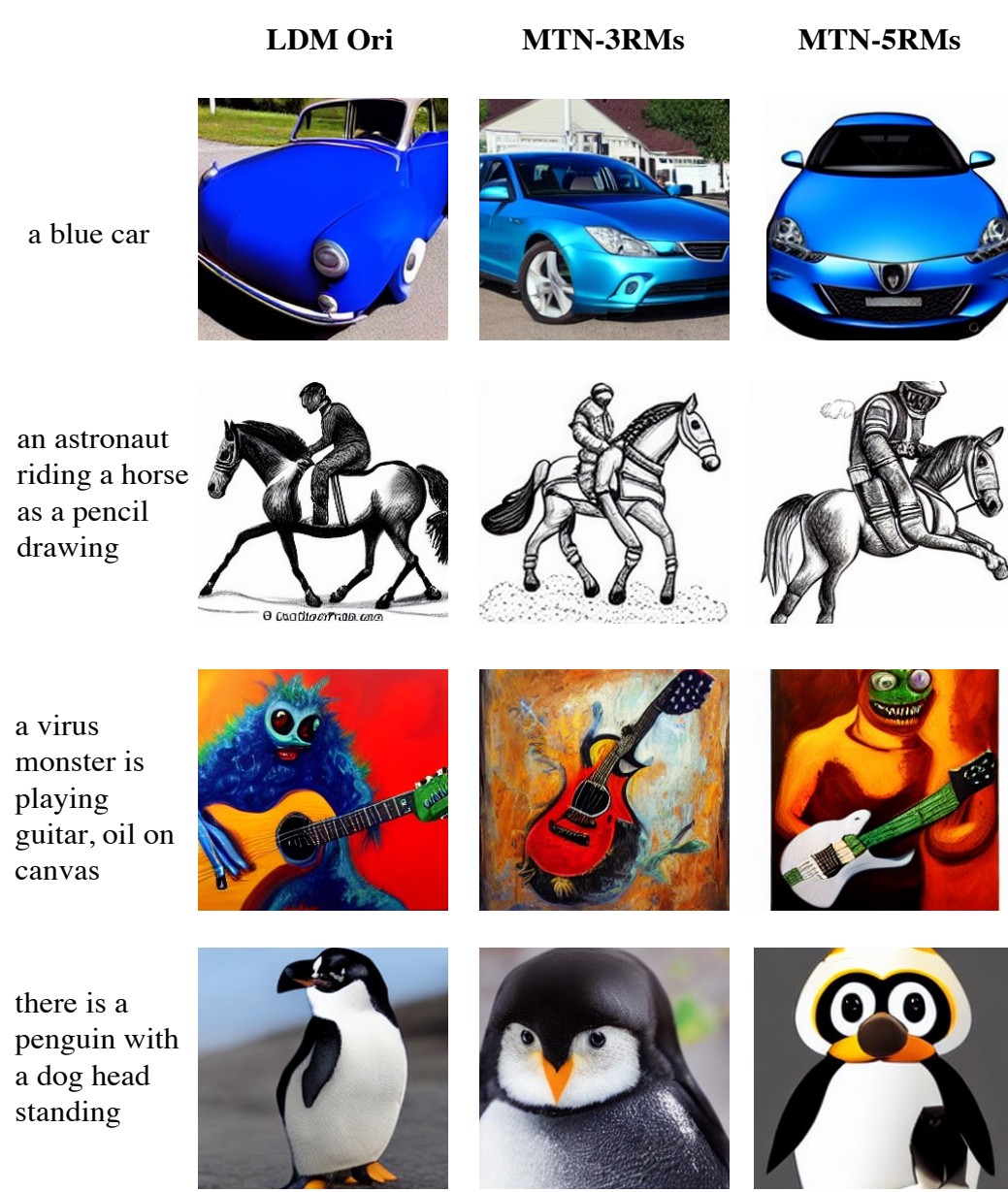

Figure 10: Monolingual generation of example prompts in $256 \times 256$ with guidance weight 7.5 and DDIM steps 200.

### A.3 ABLATION OF LOSSES

To analyze the impacts of different losses, we visualize their t-SNE map in Fig. 11. We can notice that just MSE loss cannot well align the feature of the two models. The reconstruction loss is vital to keep discrimination and avoid overfitting. The adversarial loss seems not to show much improvement in the figure. According to our empirical study, adversarial loss only brings limited gains.

We have conducted the quantitative ablation study on a subset of COCO with 5k data randomly selected. We chose T5-3B as the text encoder aligned with the LDM Unet with our MTN. FT denotes the finetuning of Unet. We have reported the FID and CLIP scores in the following table for comparison of image quality and image-text alignment. The testing data are inferred by DDIM with 200 steps. The image size is $256 \times 256$. For a comprehensive analysis, the experiments are over

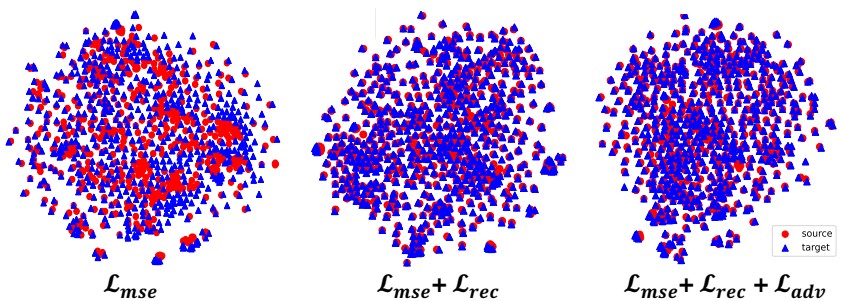

Figure 11: t-SNE (Van der Maaten & Hinton, 2008) cross-model feature map visualization.

Table 3: Ablation study of T5+MTN+LDM over the 5K COCO subset

| Ablations | | | | s=1.5 | | s=5 | | s=7.5 | |
|---|---|---|---|---|---|---|---|---|---|
| loss-mse | loss-rec | loss-adv | Finetuning | CLIP↑ | FID↓ | CLIP↑ | FID↓ | CLIP↑ | FID↓ |
| ✗ | ✗ | ✗ | ✗ | 10.70 | 98.17 | 10.75 | 141.19 | 11.00 | 156.70 |
| ✗ | ✗ | ✗ | ✓ | 18.98 | 35.76 | 22.05 | 49.14 | 22.49 | 53.11 |
| ✓ | ✗ | ✗ | ✗ | 20.40 | 34.41 | 23.01 | 48.67 | 23.40 | 52.78 |
| ✓ | ✓ | ✗ | ✗ | 20.53 | 33.14 | 23.23 | 45.96 | 23.57 | 48.67 |
| ✓ | ✓ | ✓ | ✗ | 20.67 | 32.80 | 23.24 | 45.48 | 23.74 | 48.51 |
| ✓ | ✓ | ✓ | ✓ | **21.14** | **30.93** | **23.88** | **41.92** | **24.17** | **44.58** |

three classifier-free guidances with s=1.5, 5, and 7.5 according to Eq. (2). The Tab 3 summarizes the results of our ablation study. The first row represents the direct combination of T5 and LDM, whose results do not make sense due to the severe misalignment. T5+LDM trained from scratch can be referred to in the second row. The finetuning of Unet takes 100 GPU Days, whereas MTN training needs only 5 GPU Days. Even with a ten times higher cost, its result is still inferior to those of MTNs'. By comparing the 3rd, 4th, 5th, and 6th rows, we can easily conclude the superiority of the full-version model.

### A.4    HYPER-PARAMETER ANALYSIS

More analysis of hyper-parameters is included here. The learning rate is one of the most important sensitive hyper-parameters. In Tab. 4, we explored the different learning rates to try MTN to transfer OpenClip-L to Clip-L for Stable Diffusion. It is benchmarking on the average CLIP score with 500 generative samples in the size of $512 \times 512$.

### A.5    CLOSED-LOOP CONSISTENCY

The closed-loop consistency is a helpful technique to accelerate MTN training. We have briefly introduced it in Sec 4.1 in the main paper. Here, we provide more details as illustrated in Fig. 12. The Fig. 13 and A.5 visualize the curves of alignment and reconstruction losses during training.

## B    MONOLINGUAL T2I GENERATION

### B.1    LATENT DIFFUSION MODEL

As seen in Fig. 15 and 16, we demonstrate more example results with the challenging prompts from Drawbench (Saharia et al., 2022) and Winoground (Thrush et al., 2022). The baseline methods are the original LDM and the finetuned LDM equipped with T5 as the text encoder. We can catch that the images of T5+MTN+FT are better aligned with the text and reasonable in the layout of different objects.

Table 4: Analysis of learning rate of MTN in OpenClip $\rightarrow$ CLIP.

|  | lr=1e-3 | lr=1e-4 | lr=1e-5 | lr=1e-6 |
|---|---|---|---|---|
| CLIP↑ | 27.27 | **29.53** | 27.17 | 19.40 |

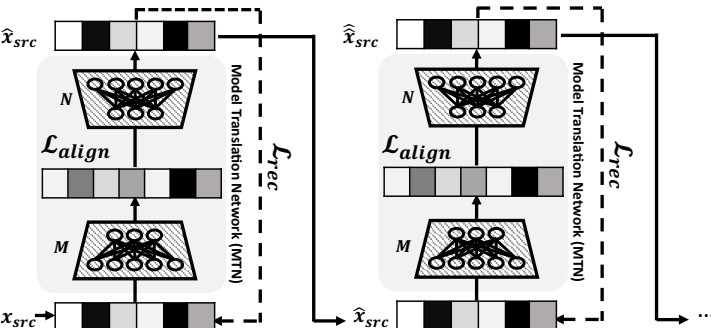

Figure 12: Illustration of Closed-loop Consistency.

## B.2 STUDY OF DIFFERENT TEXT ENCODERS

our proposed framework is generic to a wide range of text encoders. In this subsection, we have tried many different language models as the text encoder, including Bert-base (110M #params), Roberta-Large (355M #params), Clip-text (123M #params), T5-large (770M #params) and T5-3B (2.8B #params). All these pre-trained text encoders are aligned with the Latent Diffusion Model (LDM) with our proposed model to replace the original LDM text encoder without any finetuning of the LDM UNet.

These plug-and-play models are verified on a subset of COCO with 5k samples randomly selected. We have reported the FID and CLIP scores in the following table for comparison of image quality and image-text alignment. All the model translation networks (MTN) are trained by 18M sentences sampled from the captions of Laion-400M. The testing data are inferred by DDIM with 200 steps. The image size is 256×256. For a comprehensive analysis, we have conducted the experiments on three classifier-free guidance with $s$=1.5 and 5. The results are summarized in Tab. 5.

From the table, we can notice that both FID and CLIP have increased with the rise of text model sizes which is well aligned with the conclusion of Imagen(Saharia et al., 2022). The CLIP-text does not perform well here due to the larger domain gap with text-only text models such as Bert.

## B.3 MODEL TRANSFER ACROSS VARIANT TOKEN LENGTHS

MTN has a strong ability to overcome different lengths. The token number is assigned as 77, which is fixed in both LDM and SDM. Our proposed MTN can overcome the variant length text encoders without any finetuning of the Unet model. We have completed the experimental study and reported its result in the Tab. 6 to verify this point. In this experiment, we have applied Roberta-L and its tokenizers encoding maximum tokens in 77, 128, and 256, respectively. The guidance is 5, and other settings follow the Tab. 3.

## B.4 STABLE DIFFUSION MODEL

To illustrate the generics of our proposed framework, we also apply the T5-L to replace the CLIP text encoder of Stable Diffusion (v1-4). As shown in Fig. 17, our model has revealed precise control-lability and excellent visual quality compared with the standard Stable Diffusion model. Improving Stable Diffusion with efficient finetuning is our following focus.

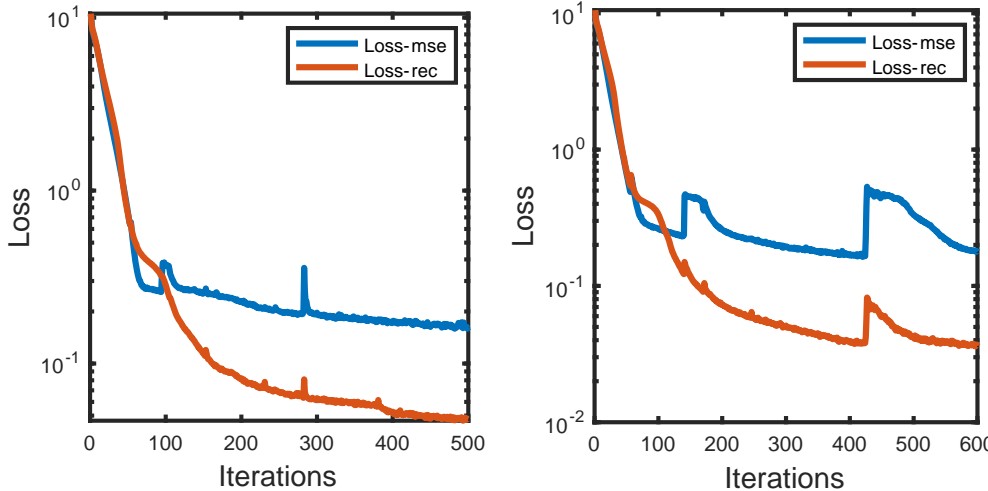

Figure 13: with Closed-loop Consistency.      Figure 14: without Closed-loop Consistency.

Table 5: Analysis of Different Text Encoders over the 5K COCO subset.

|  | Bert-B (110M) | | Roberta-L (355M) | | CLIP-Text (123M) | | T5-L (770M) | | T5-3B (2.8B) | |
|---|---|---|---|---|---|---|---|---|---|---|
|  | CLIP↑ | FID↓ | CLIP↑ | FID↓ | CLIP↑ | FID↓ | CLIP↑ | FID↓ | CLIP↑ | FID↓ |
| $s=1.5$ | 19.42 | 37.46 | 20.03 | 34.06 | 19.02 | 38.93 | 19.78 | 35.17 | **20.67** | **32.80** |
| $s=5$ | 21.97 | 55.35 | 22.85 | 45.88 | 21.85 | 47.32 | 22.76 | 47.91 | **23.24** | **45.48** |
| $s=7.5$ | 22.68 | 55.21 | 23.25 | 48.96 | 22.38 | 49.66 | 23.23 | 50.02 | **23.74** | **48.51** |

We also have applied OpenCLIP-L-14 to replace CLIP-L-14 with the similar architecture on Stable Diffusion. The quantitative evaluation is reported in Tab. 7. This is tested on COCO-5k with the guidance as 5.

## B.5 COMPUTATION COST VS PERFORMANCE

The conditional generation itself is a very challenging task where how the text encoder aligns with image Unet still needs to be discovered. We empirically find that replacing the text encoder can achieve the comparable results with the source model, but it takes a lot of effort to outperform it. The Unet may cause this as the bottleneck. In this section, we share a precise cost of computations with the different sizes of the training set in Tab. 8. This is benchmarked as the same as previous experiments on COCO-5K with T5-3B to replace the LDM text encoder with MTN. From the table, we can see that the performance has increased with the rise of data as well as the computation cost.

Moreover, we also include T5+LDM trained from scratch as a baseline. For a fair comparison, we have tried our best to train the T5+LDM with ∼140 GPU Days which is comparable with the sum of MTN116 and MTN116+FT. The results are shown in Tab. 9.

Table 6: Model Transfer with variant token lengths (SrcTokenLength →TargetTokenLength) over the 5K COCO subset with the guidance as 5. Roberta-L (Liu et al., 2019) is applied as the new text encoder to replace LDM text encoder.

|  | 77→77 | 128→77 | 256→77 |
|---|---|---|---|
| CLIP↑ | 22.85 | 23.19 | **23.37** |
| FID↓ | 45.88 | 45.67 | **45.53** |

Table 7: Transfer from OpenCLIP-L-14 (Ilharco et al., 2021) to CLIP-L-14 on Stable Diffusion.

|  | CLIP↑ | FID↓ |
|---|---|---|
| OpenCLIP+MTN+SDM | 24.66 | 36.18 |
| CLIP (Ori SDM) | **25.53** | **35.34** |

## C MULTILINGUAL T2I GENERATION

To verify the functionalities for multilingual T2I generation, we have shown more results in Fig. 18 and Fig. 19. There is one MTN for one language. To build an automatic pipeline, the model is expected to choose which MTN to use according to the detection of incoming languages.

### C.1 COMPARISON WITH MULTILINGUAL TRANSLATION MODEL

Our proposed framework is model agnostic which should be able to overcome variants domain gaps between models and languages. The cross-lingual alignment is applied as verification of proof of concept. Compared with the standard translation + DM, our proposed MTN needs less training and fewer parameters to train. For quantitative evaluation, we have compared it with a popular multilingual translation model M2M100-418M (Fan et al., 2021) (418M #params) trained by 7.5B parallel sentences. In contrast, our MTN uses far fewer training data ( 2M sentences for each pair). The results are listed in Tab. 10 where we reported the Multilingual-CLIP score [3] calculated over the 100 randomly selected multilingual image-caption pairs from Crossmodal dataset (Thapliyal et al., 2022). All the images are sampled by PLMS (Liu et al., 2022) with 50 steps in the resolution of 512 × 512.

The cosine similarity of Multilingual-CLIP text and image embeddings is reported here. From the Tab. 10, we can see that the MTN is only slightly weaker than M2M-418M despite the huge efficiency in training data. Such a result reflects the great potential of MTN in multilingual T2I by aligning the cross-lingual model with the consideration of data efficiency.

---

[3] https://huggingface.co/M-CLIP/XLM-Roberta-Large-Vit-L-14

Table 8: Analysis of MTN's training cost.

|  | Data Size | | |
|---|---|---|---|
|  | 5M | 18M | 116M |
| CLIP↑ | 20.92 | 23.24 | **23.71** |
| FID↓ | 48.63 | 45.48 | **43.17** |
| GPU Days↓ | **1.67** | 5.89 | 41.20 |

Table 9: Analysis of Performance and Computation Cost

|  | CLIP↑ | FID↓ | GPU Days↓ |
|---|---|---|---|
| LDM | 23.79 | 42.83 | - |
| T5 + FT | 23.29 | 45.62 | ∼140 |
| T5 + MTN18M | 23.24 | 45.48 | **5.89** |
| T5 + MTN18M + FT | 23.88 | 41.92 | ∼100 |
| T5 + MTN116M | 23.71 | 43.17 | 41.20 |
| T5 + MTN116M + FT | **24.14** | **41.53** | ∼100 |

Table 10: Comparison with translation model, *i.e*, M2M100-418M (Fan et al., 2021), for multilingual generation. The Multilingual-CLIP↑ score over Crossmodal (Thapliyal et al., 2022) benchmark is reported here.

|  | M2M100-418M+SDM | XLM-Roberta+MTN+SDM |
|---|---|---|
| Chinese | **24.50** | 22.01 |
| Franch | **25.08** | 23.09 |
| Spanish | **23.83** | 22.91 |
| Korean | **22.35** | 21.46 |
| Japanese | 23.73 | **23.99** |
| Ukrainian | **23.99** | 22.81 |
| Italian | 21.08 | **22.40** |

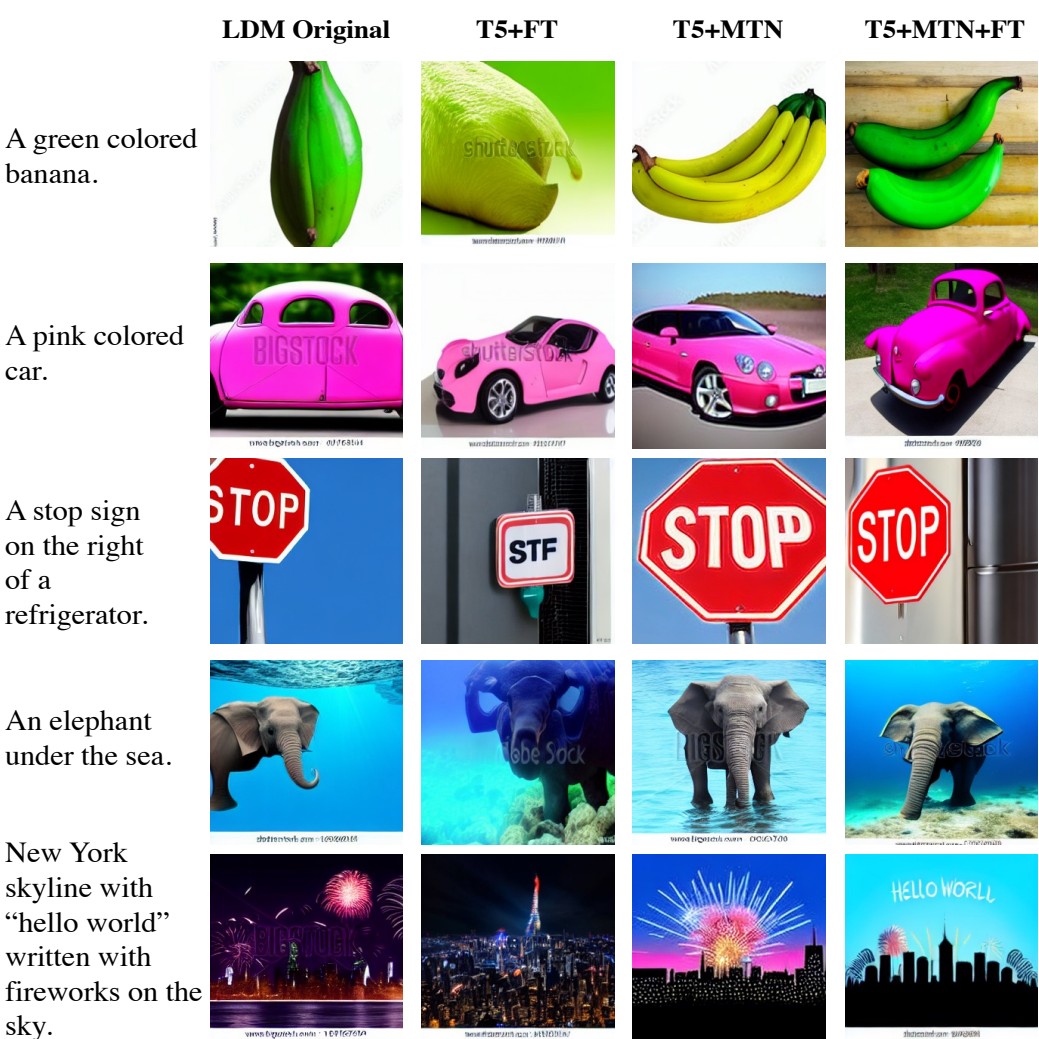

Figure 15: Monolingual generation of Drawbench prompts in 256 × 256 with guidance weight 7.5 and DDIM steps 200.

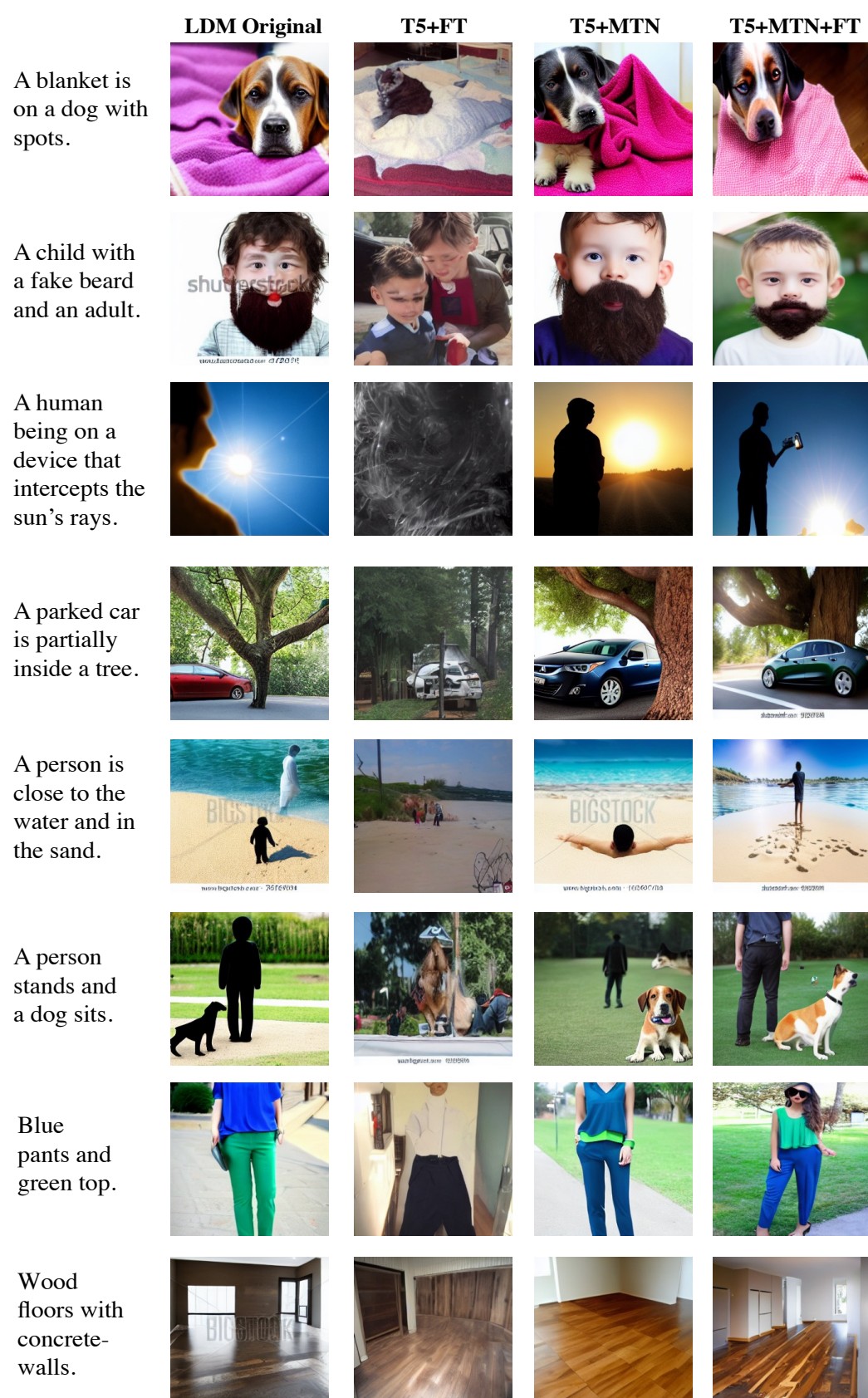

Figure 16: Monolingual generation of Winoground prompts in $256 \times 256$ with guidance weight 7.5 and DDIM steps 200.

Stable Diffusion (v1-4)          T5-L + MTN

"An astronaut
riding a horse."

"A pink car."

"A shark in the
dessert."

Figure 17: Monolingual generation of example prompts in $512 \times 512$ with guidance weight 7.5 and 50 PLMS (Liu et al., 2022) sampling steps.

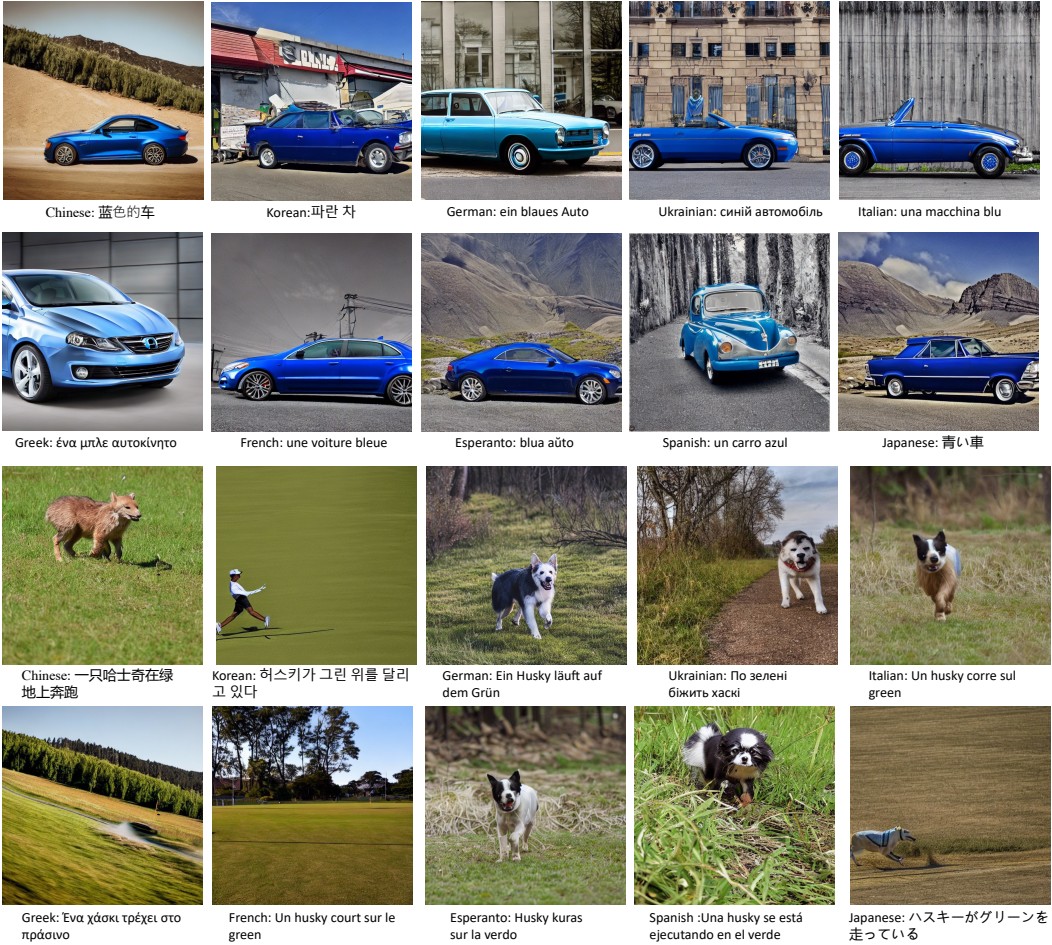

Figure 18: Multilingual generation results in 512 × 512 of XLM-Roberta + MTN + SDM decoder (sd-v1-4) with the same input captions, *i.e*, "a blue car" and "a husky running on green field", in English. It can support different languages including Japanese, Ukraine, German, Italian, Korean, Greek, Esperanto, Chinese, French and Spanish with only different MTNs. The guidance weight is 7.5 and PLMS (Liu et al., 2022) sampling steps are 50.

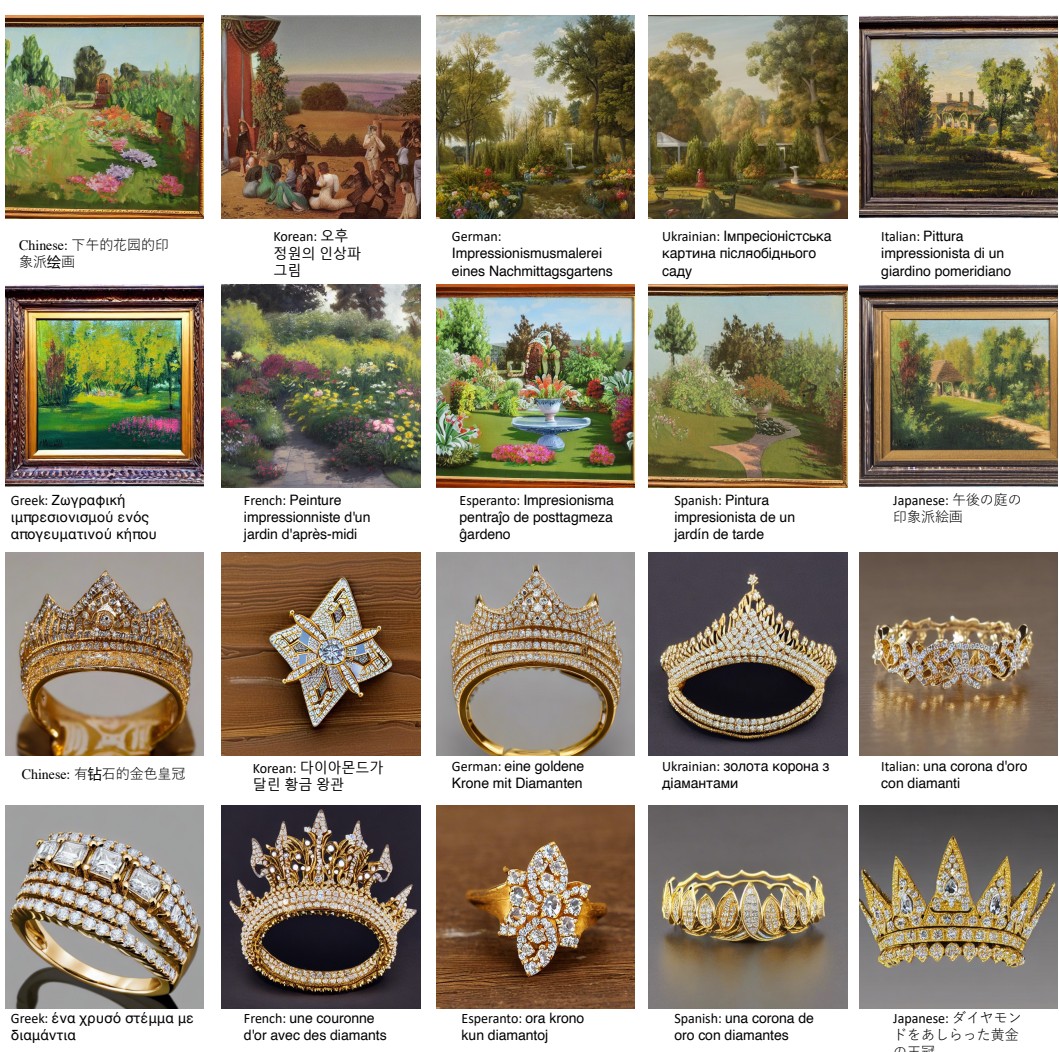

Figure 19: Multilingual generation results in $512 \times 512$ of XLM-Roberta + MTN + SDM decoder (sd-v1-4) with the same input captions, *i.e*, "Impressionism painting of a garden in afternoon" and "a golden crown with diamonds", in English.

