# OpenReview forum: "The Plug and Play of Language Models for Text-to-image Generation"
_ICLR.cc/2023/Conference — Submitted to ICLR 2023_

### Official Review · Reviewer_1TQj · 2022-10-21

**Confidence:** 3
**Correctness:** 3
**Technical Novelty And Significance:** 3
**Empirical Novelty And Significance:** 4
**Recommendation:** 6

**Clarity, Quality, Novelty And Reproducibility:**

The paper is written clearly. The proposal of switching the text encoder component of T2I without retraining is interesting and as far as I am aware, novel. The algorithm to translate between any two text encoder systems is also novel as far as I am aware.

**Details Of Ethics Concerns:**

I have no major concerns. The method, since used on T2I models, is subject to the same risks as any other T2I models.

**Strength And Weaknesses:**

My intuition divides the paper’s contribution into three major components.

1. An algorithm to translate between any two text encoder systems. The method is relatively simple and affordable, and uses three types of losses (adversarial loss, alignment loss and reconstruction loss) which are all very intuitive. The method also seems generic enough and I can imagine future applications to other domains. The writing is also very clear.

2. Applying the MTN to upgrade the BERT encoder of the LDM to the T5-3B encoder. This is an example of upgrading the monolingual text encoder of a T2I model. The result is a bit disappointing in this use case because
  - It was not until this part of the paper that I realized that end-to-end finetuning of the T2I is still needed in order to recover or improve its performance. Without the finetuning step, performance actually drops. Therefore it is not strictly “plug and play” of the new text encoder and the MTN. This is not mentioned in the paper until the experiments section, and therefore comes as a surprise and disappointment.
  - Because of the need for finetuning, it becomes a bit more complicated to argue the benefit of using the MTN. Firstly, in terms of performance, the paper compares baseline LDM, T5+FT, T5+MTN, and T5+MTN+FT, and argues that T5+MTN+FT performs better than both the baseline and T5+FT. It would be informative to also compare with LDM+T5 trained from scratch, and would be very convincing if T5+MTN+FT gets very close to that. Secondly, in terms of computational cost, it would be useful to point out the cost of training LDM from scratch, and how much cost is saved by using T5+MTN+FT vs training from scratch. (If I understand correctly, one of LDM’s advantages is training affordability, which means MTN may not be saving much. It would be nice to show this is not the case.)

3. Applying the MTN to upgrade the CLIP text encoder of SDM to the multilingual XLM-Roberta-L. This is a successful application. The hybrid text inputs are interesting as well, but not that surprising. I think it is more contributed by the capability of XLM-Roberta-L than the MTN, since XLM-Roberta-L aligns representations across languages very well, and therefore is able to handle hybrid text encoding as well. One disadvantage is that it requires parallel corpus to train the MTN, whereas retraining SDM with XLM-Roberta-L does not require extra data and should also automatically work with multilingual prompts.


**Questions:**
- In the user study using Amazon Turk (Fig 4), were images from the three models displayed side by side, and the Turk picks the best out of three? If that is the case, I am not sure the results (31.7%, 32.7%, 35.7%) significantly deviate from chance statistically.
- What is the dataset used for finetuning T5+FT and T5+MTN+FT?
- Is the text encoder frozen during finetuning T5+FT and T5+MTN+FT?
- What’s the performance for LDM retrained with T5? How does it compare with T5+MTN+FT?
- What's the compute comparison between LDM retrained with T5 and T5+MTN+FT?


**Summary Of The Paper:**

This paper proposes a method that allows replacing the text encoder in a text-to-image model without retraining the entire model. It uses a Model Translation Network (MTN) that is trained with text corpus data (or parallel text data for multilingual text encoders) to align the representation spaces of two text encoders. The paper demonstrates two application use cases: to update the text encoder of an existing T2I model to a more powerful one, and to update the text encoder of a monolingual T2I model to a multilingual one.

**Summary Of The Review:**

Overall I like the idea proposed in the paper. I think it is important to think about how to reuse trained models and components, as in many cases it is unaffordable or wasteful to retrain end-to-end models. The proposed algorithm to align two text encoders’ representations is simple and clear. The main drawback for me is that I am not convinced of its benefit in upgrading the text encoder in a T2I model, as shown by the results in Section 5.2. We may need better results, or better-organized results, to justify the claim that it “boosts the performance of existing T2I models”. Therefore I think it is marginally below the acceptance threshold. I am willing to change my score if either the claims or the results are adjusted to align better with each other.

---

> ### Author Response · Authors · 2022-11-19
> **Response to Reviewer 1TQj (Part 1 of 2)**
>
> Thanks for your valuable time reviewing this paper. Your questions and comments are all very constructive. We provide our responses below, and we have updated our paper manuscript with changes highlighted in red. Please feel free to let us know if you have additional questions and we are always glad to discuss them.
>
> **Q1: End-to-end finetuning of the T2I is still needed.**
>
> Thank you for this comment. The plug-and-play of text models in the T2I problem is essentially different from other multimodal tasks, such as captioning. The conditional generation itself is a very challenging task where how the text encoder aligns with image Unet still needs to be discovered. We empirically find that replacing the text encoder can achieve the comparable results with the source model, but it takes a lot of effort to outperform it. The Unet may cause this as the bottleneck. We have conducted the ablation analysis with different sizes of training sets for MTN. The results are shown as follows:
>
> COCO-5k, guidance=5
>
> |  | CLIP | FID |GPU Days (A100)|
> | --- | --- | --- |--- |
> | Ori LDM |23.79  |42.83  |-|
> |  T5+MTN18M| 23.24 | 45.48 |5.89|
> |  T5+MTN18M+FT| 23.88 | 41.92 |~100|
> |  T5+MTN116M|  23.71| 43.17 |41.20|
> |  T5+MTN116M+FT| **24.14** |**41.53**  |~100|
>
> The table above indicates that the MTN can be improved with more training data (MTN18M indicates using 18M sentences for training). This also reveals the trade-off between performance and cost, where we can sacrifice one for the other. Sec. B. 5 has been updated to include the discussion.
>
>
> **Q2: Comparison with Training T5+LDM from Scratch. What’s the performance for LDM retrained with T5? How does it compare with T5+MTN+FT? What's the compute comparison between LDM retrained with T5 and T5+MTN+FT?**
>
> Thank you for the question of T5 finetuning. We agreed that T5+LDM trained from scratch is an essential baseline to be compared. However, this is very time costly due to the upgrade of text encoders where the T5+LDM Unet is almost two times larger than the original LDM. Therefore, it needs more computation to fulfill. For a fair comparison, we have tried our best to train the T5+LDM with ~140 GPU Days which is comparable with the sum of  MTN116 and MTN116+FT. The results are listed in the following table.
>
> COCO-5k, guidance=5
>
> |  | CLIP | FID |GPU Days (A100)|
> | --- | --- | --- |--- |
> |T5+LDM+FT| 22.05 | 49.19|~100  |
> |T5+LDM+FT| 23.29 | 45.62|~140  |
> |  T5+MTN116M|  23.71| 43.17 |41.20|
> |  T5+MTN116M+FT| **24.14** |**41.53**  |~100|
>
>
> From the table, we can see that the T5+LDM is inferior of the MTN-based model in both performance and training efficiency.
>
> The T5+LDM is still not converged, which can be improved further with more training. But this is out of our capacity. We are uncertain of how much computation is needed for LDM training, and the authors of LDM do not provide the official code of T2I model training on Laion-400M (they only shared the T2I inference code and the training code of class-to-image generation on ImageNet). Our implementation is based on the ImageNet code, which may be different from the official training pipeline on Laion-400M. However, all our reported experiments are conducted under the same setting to be fairly compared.
>
>
>
> **Q3: Training XLM-Roberta + SDM from scratch**
>
> Thank you for this question. The multilingual generation without translation needs huge data and computation for training. We have tried to train the XLM-Roberta + SDM with Laion-2B multilingual dataset. But this is extremely slow since we have only limited GPUs. After a few days of training (~100 GPU Days), the first epoch is still not finished, and we have not received reasonable visual results given multilingual prompts. Such a task may need comparable computation cost as training SDM from scratch (thousands of GPU days). Therefore, some acceleration techniques should be necessary for this task, but it is beyond the scope of this paper.

---

> > ### Comment · Reviewer_1TQj · 2022-11-26
> > **Thank you for the response**
> >
> > Thank you for the detailed response. My main concern was how much benefit this method brings, with its complication of training the MTN first, and then finetuning the entire model, for a somewhat small improvement in FID. The authors' rebuttal showed that even with the above two steps, it still saves cost significantly compared to training from scratch. I am more positive than my initial score, and now think the paper can be marginally above the threshold.

---

> ### Author Response · Authors · 2022-11-19
> **Response to Reviewer 1TQj (Part 2 of 2)**
>
> **Q4: Question of Amazon Turk.**
>
> Thank you for this question. Yes, we have presented multiple generation results in a window and asked the Turk to select the best. Specifically, we have applied 993 generated triplets for user study with three repeats to avoid randomness. During each repeat, the data order will be randomly shuffled. Therefore, there are 2979 polls in total. Our method has won 35.7% of votes whose p-value is **0.04** computed by the *scipy.stats* package (https://docs.scipy.org/doc/scipy/reference/generated/scipy.stats.binomtest.html).
> This is smaller than 0.05 and can be regarded as statistically significant. Sec. 5.2.1 has been updated to report the p-value accordingly.
>
>
> **Q5: Dataset used for finetuning T5+FT and T5+MTN+FT.**
>
> Thank you for this question. We have selected a subset of 116M image-text pairs from Laion-400M filtered by the BLIP [1]. All these data are used for model finetuning.
>
> [1] BLIP: Bootstrapping Language-Image Pre-training for Unified Vision-Language Understanding and Generation. ICML 22.
>
> **Q6: Is the text encoder frozen during finetuning T5+FT and T5+MTN+FT?**
>
> Yes. We have fixed the parameters of the text part during finetuning.

---

### Official Review · Reviewer_2Z7M · 2022-10-24

**Confidence:** 3
**Correctness:** 3
**Technical Novelty And Significance:** 3
**Empirical Novelty And Significance:** 2
**Recommendation:** 6

**Clarity, Quality, Novelty And Reproducibility:**

**Clarity**

The writing and the presentation are clear. If the author can provide more clear mathematical notation, in Section 4.1, it will be much better, e.g., some problem setups in a mathematical formula.

**Quality**

The overall quality is fine.

**Novelty**

The method itself is a combination of existing works, but the research direction is novel.

**Reproducibility**

Since the authors did not provide the code and the hyper-parameter, I believe it is hard to be reproduced.

**Strength And Weaknesses:**

**Strength**

(1) The research direction itself is quite interesting, i.e., replacing the text encoder for a better (or multilingual) generative model.

(2) The presentation and organization are clear.

(3) The author provides comprehensive generated results (in the Supplementary).

----
**Weakness**

(1) Ablation study is missing. Only the quantitative results are given in Appendix (A.3). To claim the effectiveness of each component, the qualitative results should be given.

(2) The choice of using domain adaptation is somewhat unclear. Since authors have to run an additional discriminator, it may increase the computation, which is slightly against the motivation. Does this bring a significant improvement?

(3) The performance gain is somewhat marginal (in Table 1). Since the LDM is not properly reproduced, the gain seems to be about -0.5 FID or -1.5 FID, which is slightly marginal. Can the author try a bigger language model to achieve more gain? E.g., providing encoder size and FID results like in Figure 4 of [1].

(4) Comparison with a naive approach for a multilingual generation. One can simply use a state-of-the-art machine translation model (e.g., CONT [2]) on top of LDM (or SDM) and fine-tune it. Does the proposed approach outperform such a naive approach?

(5) (minor) Some analysis of the hyper-parameter will be great (e.g., hyper-parameter sensitivity)

[1] Saharia et al., Photorealistic Text-to-Image Diffusion Models with Deep Language Understanding, NeurIPS 2022\
[2] An et al., CONT: Contrastive Neural Text Generation, NeurIPS 2022

**Summary Of The Paper:**

The paper aims to efficiently replace the text encoder of the text-to-image generation model to improve the generation quality (or to make a multilingual text-to-image generative model). To this end, the authors proposed Model Translation Network (MTN) in which training loss is motivated by domain adaptation and cross-domain alignment. The proposed method shows that the performance gain is consistent over the recent latent diffusion model (LDM) and can make LDM a multilingual text-to-image generative model.

**Summary Of The Review:**

I recommend weak acceptance. I believe the target problem is quite interesting and has the strength to be accepted. However, some concerns (mentioned in the weakness part) make me choose the weak acceptance.

---

> ### Author Response · Authors · 2022-11-19
> **Response to Reviewer 2z7M (Part 1 of 2)**
>
> Thank you for your constructive comments and questions. We provide our responses below, and we have revised our paper manuscript with changes highlighted in red. Please let us know if you have additional comments. We will always be happy to discuss.
>
>
> **Q1: Ablation study is missing.**
>
> Thank you for pointing this out. We agree that more ablation analysis of our proposed method is needed. During rebuttal, we conducted a comprehensive quantitative evaluation on a subset of COCO with 5k data randomly selected (more testing data will be too costly for us). We chose T5-3B as the text encoder aligned with the LDM Unet by our proposed MTN. FT denotes the Unet finetuning. The ablation results of different loss terms can be referred to in the following table.
>
> | L_{mse}| L_{rec}|L_{adv} | FT| s=1.5, CLIP |  s=1.5, FID | s=5, CLIP |  s=5, FID |  s=7.5, CLIP |  s=7.5, FID |
> | --- | --- | --- | --- | --- | --- |--- | --- | --- | --- |
> |x  | x |  x| x|  10.70| 98.17 |10.75 | 141.19|11.00 |  156.70|
> |x  | x | x| ✓| 18.98 |35.76  |22.05 | 49.14|22.49  |53.11  |
> |✓  | x | x| x| 20.40 | 34.41 | 23.01|48.67 | 23.40 | 52.78 |
> |✓  | ✓ | x| x| 20.53 | 33.14 | 23.23| 45.96  | 23.57 | 48.67|
> |✓  | ✓ | ✓| x| 20.67 |32.80  |23.24 |45.48 | 23.74 | 48.51 |
> |✓  | ✓ | ✓| ✓| **21.14** | **30.93** | **23.88**| **41.92**| **24.17** | **44.58** |
>
> The first row describes the simple combination of T5 and LDM, which has severe misalignment with a very poor FID and CLIP score, indicating a nonsense result. T5+LDM trained from scratch can be referred to in the second row. The finetuning of Unet takes ~100 GPU Days, whereas MTN training needs only 5 GPU Days (A100). Even with 10x computation cost, its result is still inferior to those of MTNs'. By comparing the third, fourth, fifth, and sixth rows, we can easily notice the superiority of our full-version model. Sec. A. 3 has been updated to include this part.
>
>
> **Q2: The choice of using domain adaptation is somewhat unclear. It may increase the computation.**
>
> Thank you for your comment. The adversarial loss is designed as an auxiliary objective to help distribution-wise alignment upon the direct matching loss. We have implemented the discriminator as a simple MLP model with less than 1M parameters. Therefore, it is very efficient and only increases ~10% of computations overall. Even though the L_{adv} is not significant for performance boosting, we can still leave it considering the marginal computation cost.
>
> **Q3: Performance gain is somewhat marginal.**
>
> Thank you for this comment. The plug-and-play of text models in the T2I problem is essentially different from other multimodal tasks, such as captioning. The conditional generation itself is a very challenging task where how the text encoder aligns with image Unet still needs to be discovered. We empirically find that replacing the text encoder can achieve comparable results with the source model, but it takes a lot of effort to outperform it. The Unet may cause this as the bottleneck. We have conducted the ablation analysis with different sizes of training sets for MTN. The results are shown as follows:
>
> COCO-5k, guidance=5
>
> |  | CLIP | FID |GPU Days (A100)|
> | --- | --- | --- |--- |
> | Ori LDM |23.79  |42.83  |-|
> |  T5+MTN18M| 23.24 | 45.48 |5.89|
> |  T5+MTN18M+FT| 23.88 | 41.92 |~100|
> |  T5+MTN116M|  23.71| 43.17 |41.20|
> |  T5+MTN116M+FT| **24.14** |**41.53**  |~100|
>
>
>
> For the user study on Amazon Turk , we have generated 993 triplets with three repeats to avoid randomness. During each repeat, the data order will be randomly shuffled. Therefore, there are 2979 polls in total. Our method has won 35.7% of votes in controllability whose p-value is **0.04**. It can be regarded as statistically significant.

---

> > ### Comment · Reviewer_2Z7M · 2022-12-02
> > **Reponse**
> >
> > Thank you for your efforts in answering my questions.
> >
> > While there is one concerning point, I am satisfied with the reply and will keep my positive score. I believe the paper has the strength to be accepted.
> > - concern point: the domain adaptation loss does not increase the performance much but increases the computation by about 10% (note that such training requires quite a heavy computation, so 10% is not very small).
> >
> > Best,
> > Reviewer 2z7M

---

> > > ### Author Response · Authors · 2022-12-02
> > > **Follow-up Response**
> > >
> > > Dear Reviewer 2z7M,
> > >
> > > Thank you for your positive recognition of our submission. We sincerely appreciate your comments and suggestions for this paper. As for the computation of adversarial loss, we may remind that the MTN training is not very costly compared with the post-finetuning. The ~7 GPU days are enough mostly. Therefore, its 10% would not cause much increase in the budget.
> > >
> > > Thanks,
> > > Authors of Paper5567

---

> ### Author Response · Authors · 2022-11-19
> **Response to Reviewer 2z7M (Part 2 of 2)**
>
> **Q4: Can the author try a bigger language model to achieve more gain?**
>
> Thank you for this question. The Imagen paper has discovered that increasing the sizes of text models is critical for T2I, which can be referred to in its Fig. 4 [1]. Our paper attempts to explore the plug-and-play of different text encoders for existing T2I models. Due to the limits of our computation capacity, we have verified this point by comparing a series of text encoders, including Bert-base (110M #params), Roberta-large (355M #params),  T5-large (770M #params) and T5-3B (2.8B #params). This is benchmarked on a subset of COCO with 5k samples randomly selected. We have reported the FID and CLIP scores in the following table for comparison of image quality and image-text alignment. All the model translation networks (MTN) are trained by 18M English sentences. The testing data are generated by DDIM with 200 steps. The image size is 256*256.
>
> guidance = 1.5
>
> | | CLIP | FID|
> | --- | --- | --- |
> |  Bert-base| 19.42 |37.46  |
> |  Roberta-large| 20.03 |34.06  |
> |  T5-Large| 19.78 | 35.17 |
> |  T5-3B|**20.67** | **32.80** |
>
> guidance = 5
>
> | | CLIP  | FID|
> | --- | --- | --- |
> |  Bert-base| 21.97 |55.35  |
> |  Roberta-large| 22.85 | 45.88 |
> |  T5-Large|22.76  |47.91  |
> |  T5-3B|**23.24**  |  **45.48**|
>
> guidance = 7.5
>
> | | CLIP  | FID|
> | --- | --- | --- |
> |  Bert-base| 22.68 |55.21  |
> |  Roberta-large| 23.25| 48,96 |
> |  T5-Large| 23.23  | 50.02  |
> |  T5-3B|**23.74**  |  **48.51**|
>
> From the table, we can notice that both FID and CLIP have increased with the rise of text model sizes which is well aligned with the conclusion of Imagen[1]. The high guidance score could decrease the FID due to the trade-off between image quality and controllability. Sec. B. 2 has been updated to include this part.
>
>
> [1] Photorealistic Text-to-Image Diffusion Models with Deep Language Understanding. NeurIPS 22.
>
>
>
>
>
> **Q5: Comparison with a naive approach for a multilingual generation?**
>
>
> Thank you for this suggestion. Our proposed framework is model agnostic which should be able to overcome variants domain gaps between models and languages. The cross-lingual alignment is applied as a proof of concept for such a broad capacity. Compared with the standard translation + DM, our proposed MTN needs less training and fewer parameters to train. For quantitative evaluation,  we have compared it with a popular multilingual translation model M2M_418M [2] (418M #params) trained by 7.5B parallel sentences, whereas our MTN uses far less training data (~2M sentences for each pair). The results are shown below (guidance = 7.5, 50 PLMS steps):
>
> |  | Chinese | Franch | Spanish | Korean |Japanese|Ukrainian|Italian|
> | --- | --- | --- |--- | --- | --- |--- | --- |
> | M2M_418M+SDM |**24.50**|**25.08**| **23.83** | **22.35** |23.73|**23.99**|21.08|
> | MTN+SDM|22.01  |23.09  | 22.91 |21.46  |**23.99**|22.81|**22.40**|
>
> We used 100 randomly selected multilingual captions from Crossmodal-3600 [3] as the benchmark. The cosine similarity of Multilingual-CLIP between text and image embeddings is reported here. From the table, we can see that the MTN is only slightly weaker than M2M_418M despite the huge efficiency in training data. Such a result reflects the great potential of MTN in multilingual T2I by aligning the cross-lingual model with the consideration of data efficiency. Sec. C. 1 has been updated to have this part.
>
>
> [2] Beyond English-Centric Multilingual Machine Translation. JMLR 21.
>
> [3] Crossmodal-3600: A Massively Multilingual Multimodal Evaluation Dataset. EMNLP 22.
>
> **Q6: Analysis of the hyper-parameter.**
>
>
> Thank you for pointing this out. We have followed your suggestion to include more analysis of  hyper-parameters. The learning rate is one of the most important sensitive hyper-parameters. We explored the different learning rates to try MTN to transfer OpenClip-L to Clip-L for Stable Diffusion. It is benchmarking on the average CLIP score with 500 generative samples in the size of 512 * 512 (guidance = 5, 50 PLMS steps).
>
> | | lr=1e-3 |  lr=1e-4  |  lr=1e-5  |  lr=1e-6  |
> | --- | --- | --- |--- | --- |
> | OpenCLIP+MTN+SDM  |  27.27  |  **29.53**| 27.17 |  19.40|
>
> Therefore, we assign the learning rate as 1e-4 as the default setting. We will include more hyper-parameters in the later version manuscript (Sec. A.4).
>
> **Q7: Revision of mathematical notation, in Section 4.1. Problem setups in a mathematical formula.**
>
> We greatly appreciate this suggestion that more mathematical notations and formulas are needed to systematically illustrate our task. We have followed your comments to revise Sec 4.1 in the latest manuscript (highlighted in red).

---

### Official Review · Reviewer_5yb4 · 2022-10-24

**Confidence:** 4
**Correctness:** 3
**Technical Novelty And Significance:** 3
**Empirical Novelty And Significance:** 3
**Recommendation:** 6

**Clarity, Quality, Novelty And Reproducibility:**

The paper is a clear one. However it suffers issues in quality and novelty as mentioned above.


**Strength And Weaknesses:**

# Strength

The problem this paper addresses is a novel and very interesting one per se, and is well-motivated. If fully fulfilled, this would lead to a large number of applications.

The authors make a reasonable effort to show the performance of this method. The effort in qualitative experiments is great in that lots of examples are shown for comparison. The method would be welcomed by the community.

# Weakness

With the strength said, I'm not fully convinced of the proposed method, as detailed below:

Proposed algorithm for alignment: The proposed method projects the new text-encoder's output space to the old encoder's space, thus can serving as a wrapper over new text encoder. However, several issues are presenting:

1. The training loss, which consists of three losses (eq 3,4,5), are complex and the dynamic is unclear. No ablation study is presented to provide the justification of such a design.
2. It is unclear if the architecture of MTN could handle fixed length representation or the variant length ones.
3. The technical contribution of the proposed MTN is unclear. Related works in bridging the latent space of different language models are not discussed. Just to name a few, in machine translation:
    - Kulshreshtha et al. Cross-lingual Alignment Methods for Multilingual BERT: A Comparative Study
    - Cao et al. Multilingual Alignment of Contextual Word Representations
4. The performance contribution, or how good the MTN works, is unclear. MSE loss seems to be okay as shown in the experiment, however there is not a grounding of exactly how well MTN works.  We have no idea if the qualitative results come from the MTN being just okay but the generative model for images is powerful enough to compensate for MTN. Ideally, a good ablation is to have two runs of the same language models with different initial parameters and using the proposed technique to replace one with another.
5. One argument for this work is that a new language model's ability could be leveraged. However if it's demonstrated through an language model in another language, a justification should be provided through comparing it with simply translating the prompt.

# Questions for Authors

1. How much computation is required for the proposed MTN?


**Summary Of The Paper:**

In this paper, the authors proposed to address the problem of replacing a text encoder in a text-to-image model without re-training the whole pipeline from scratch. In doing so, it is proposed that MTN (Model Translation Network) projects the output of the new text encoder to the old one's output space, enabling the replacement. Experiments show both qualitatively and quantitatively the proposed baseline is capable of doing a replacement.


**Summary Of The Review:**

While this paper addresses an interesting problem and is well-motivated, the issues in the proposed method and the experiment's design makes it hard to be presented in the conference in its current form. I would thus suggest a rejection of this manuscript based on the current understanding, but I'm happy to change my rating if my concerns and questions are addressed in the discussion.

---

> ### Author Response · Authors · 2022-11-19
> **Response to Reviewer 5yb4 (Part 1 of 2)**
>
> Thank you very much for the valuable comments. We provide our responses below, and we have revised our paper where the changes are highlighted in red. Please feel free to let us know if you have additional comments and we are always glad to discuss.
>
> **Q1: The training loss is complex and unclear. No ablation study is presented to provide the justification of such a design.**
>
> Thank you for this question. The motivation for three loss terms is discussed in Sec. 4.1, and the visualization analysis is provided in Fig. 11 in Appendix. Despite these explorations, we agree that we need more analysis to evaluate the influences of these loss terms, which could help us understand its mechanism deeper. We have conducted the quantitative evaluation on a subset of COCO with 5k data randomly selected (more testing data will be too costly for us). We choose T5-3B as the text encoder aligned with the LDM Unet with our MTN. FT denotes the finetuning of Unet. The following table summarizes the results of our ablation study.
>
> | L_{mse}| L_{rec}|L_{adv} | FT| s=1.5, CLIP |  s=1.5, FID | s=5, CLIP |  s=5, FID |  s=7.5, CLIP |  s=7.5, FID |
> | --- | --- | --- | --- | --- | --- |--- | --- | --- | --- |
> |x  | x |  x| x|  10.70| 98.17 |10.75 | 141.19|11.00 |  156.70|
> |x  | x | x| ✓| 18.98 |35.76  |22.05 | 49.14|22.49  |53.11  |
> |✓  | x | x| x| 20.40 | 34.41 | 23.01|48.67 | 23.40 | 52.78 |
> |✓  | ✓ | x| x| 20.53 | 33.14 | 23.23| 45.96  | 23.57 | 48.67|
> |✓  | ✓ | ✓| x| 20.67 |32.80  |23.24 |45.48 | 23.74 | 48.51 |
> |✓  | ✓ | ✓| ✓| **21.14** | **30.93** | **23.88**| **41.92**| **24.17** | **44.58** |
>
> The first row represents the direct combination of T5 and LDM, whose results do not make sense due to the severe misalignment. T5+LDM trained from scratch can be referred to in the second row. The finetuning of Unet takes 100 GPU Days, whereas MTN training needs only 5 GPU Days. Even with a ten times higher cost, its result is still inferior to those of MTNs'. By comparing the 3rd, 4th, 5th, and 6th rows, we can easily conclude the superiority of the full-version model. Sec. A. 3 has been revised to include these results and discussions.
>
> **Q2: It is unclear if the architecture of MTN could handle fixed length representation or the variant length ones.**
>
> Thank you for mentioning variant length. Yes, the MTN has a strong ability to overcome different lengths. First of all, the text models such as T5 or Bert can project the different lengthening captions into a sequence of tokens, which, therefore, enables the Unet to generate the image from different lengths of text. The number of token is set as 77, which is fixed for both LDM and SDM. Our proposed MTN can overcome the variant length text encoders without any finetuning of the Unet model. We have completed the experimental study and reported its result in the following table to verify this point.
>
>
>
> | | MaxTokLength=77 | MaxTokLength=128 |MaxTokLength=256 |
> | --- | --- | --- |--- |
> |  CLIP| 22.85 | 23.19 | **23.37**  |
> |FID  | 45.88 | 45.67 |**45.53** |
>
> In this experiment, we have applied Roberta-L and its tokenizers encoding the maximum tokens in 77, 128, and 256, respectively. The guidance is 5, and other settings follow the previous experiment (guidance = 5). Sec. B. 3 has been updated to include this part.
>
> **Q3: The technical contribution of the proposed MTN is unclear. Comprison with Cross-lingual Alignment Methods[1][2].**
>
> Thank you for this question. The cross-lingual alignment is a well-explored topic in the area of multilingual language models. These methods attempt to align the representation of cross-lingual inputs based on a multilingual text model, such as Multilingual Bert. [1] has comprehensively explored the different alignment methods, including rotation and finetuning alignment. [2] has modeled the contextual token-wise embeddings to aggregate richer information for cross-lingual alignment.
>
> Our paper considers another problem in aligning different text models with the same or parallel data. This is essentially different from cross-lingual alignment, with more complicated and challenging mismatches across the source and target. In specific, except for the token-wise and contextual alignment, our proposed MTN can also overcome the feature dimension change (LDM-Text with 1280-dim -> T5 with 1024-dim) and variant token length (LDM-Text with 77 tokens -> Roberta with 256 tokens), which, however, cannot be addressed by the cross-lingual alignment methods. The latest manuscript has included a discussion of cross-lingual alignment (Sec. 2.2), which is very helpful for us. We will continue to explore this direction in the future.
>
>
> [1] Cross-lingual Alignment Methods for Multilingual BERT: A Comparative Study. EMNLP 20.
>
> [2] MULTILINGUAL ALIGNMENT OF CONTEXTUAL WORD REPRESENTATIONS. ICLR 20.

---

> ### Author Response · Authors · 2022-11-19
> **Response to Reviewer 5yb4 (Part 2 of 2)**
>
> **Q4: The performance contribution, or how good the MTN works, is unclear. We have no idea if the qualitative results come from the MTN being just okay but the generative model for images is powerful enough to compensate for MTN.**
>
> Thank you for pointing this out. This is a very important question, and we have tried more quantitative evaluations to verify the effectiveness of MTN. All the benchmarking settings follow the experiment presented in A1.
>
> First of all, we have conducted experiments on LDM with Bert-base (110M #params), Bert-large (340M #params), and Roberta-large (355M #params) with fewer parameters than the LDM-text (581M #params). The guidance is 5.
>
> | | CLIP  | FID|
> | --- | --- | --- |
> |  Bert-base+MTN+LDM| 21.97 |55.35  |
> |  Bert-large+MTN+LDM| 22.81 |46.15  |
> |  Roberta-large+MTN+LDM| 22.85 | 45.88 |
> | Ori LDM| **23.79** | **42.83** |
>
> Similarly, we have applied OpenCLIP-L-14 to replace CLIP-L-14 with the similar architecture on Stable Diffusion (guidance = 5).
>
>
>
> | | CLIP  | FID|
> | --- | --- | --- |
> |  OpenClip+MTN+SDM|24.66  |36.18|
> | CLIP (Ori SDM) |**25.53**|**35.34** |
>
>
> The above results support the stability of MTN in aligning different text models with Unet. Sec. B. 5 has been updated to include this part.
>
>
> **Q5: Comparison with simply translating the prompt on top of T2I model**
>
>
> Thank you for this suggestion. Our proposed framework is model agnostic which should be able to overcome variants domain gaps between models and languages. The cross-lingual alignment is applied as a proof of concept for such a broad capacity. Compared with the standard translation + DM, our proposed MTN needs less training and fewer parameters to train. For quantitative evaluation,  we have compared it with a popular multilingual translation model M2M_418M [3] (418M #params) trained by 7.5B parallel sentences, whereas our MTN uses far less training data (~2M sentences for each pair). The results are shown below:
>
> |  | Chinese | Franch | Spanish | Korean |Japanese|Ukrainian|Italian|
> | --- | --- | --- |--- | --- | --- |--- | --- |
> | M2M_418M+SDM |**24.50**|**25.08**| **23.83** | **22.35** |23.73|**23.99**|21.08|
> | MTN+SDM|22.01  |23.09  | 22.91 |21.46  |**23.99**|22.81|**22.40**|
>
> We used 100 randomly selected multilingual captions from Crossmodal-3600 [4] as the benchmark. The cosine similarity of Multilingual-CLIP text and image embeddings is reported here. From the table, we can see that the MTN is only slightly weaker than M2M_418M despite the huge efficiency in training data. Such a result reflects the great potential of MTN in multilingual T2I by aligning the cross-lingual model with the consideration of data efficiency. Sec. C. 1 has been updated accordingly.
>
>
> [3] Beyond English-Centric Multilingual Machine Translation. JMLR 21.
>
> [4] Crossmodal-3600: A Massively Multilingual Multimodal Evaluation Dataset. EMNLP 22.
>
>
>
>
> **Q6: How much computation is required for the proposed MTN**
>
> Thank you for this question. In Sec 5.1, we have introduced the rough range of MTN training on A100 GPU. Here we share a precise cost of computations with the different sizes of the training set. This is benchmarked on a subset of COCO with 5k samples. The images are synthesized by DDIM in 200 steps (guidance = 5).
>
> | | 5M | 18M |116M |
> | --- | --- | --- |--- |
> | CLIP |20.92  | 23.24 |**23.71**|
> |  FID| 48.63 |45.48  |**43.17**|
> |  GPU Days (A100)|**1.67**  |5.89  | 41.20|
>
> From the table, we can see that the performance has increased with the rise of data as well as the computation cost. Sec. B. 5 has been updated to include this.

---

> ### Comment · Reviewer_5yb4 · 2022-11-25
> **Reply to authors's response**
>
> I thank the authors for the substantial discussion and revision which answers my questions. I believe some of my concerns are addresses, especially on the ablation study, further comparison, and the computation cost.
>
> However I believe hat my concern raised on Q4 is not fully addressed, especially the two-run setup that could show an upper bound of the proposed method.
>
> That said I would highly appreciate the effort and I am more possible about the paper now. I increase my score to 6.

---

> > ### Author Response · Authors · 2022-11-25
> > **Follow-up Response**
> >
> > Dear Reviewer 5yb4,
> >
> > Thank you very much for your response, and we really appreciate your recognition of this paper. But we are not sure if there are some errors with our web since we have seen a change of the score in the system. As for Q4, could you mind sharing more details of the suggested experiments that would be very helpful for paper quality enhancement.
> >
> > Thank you!
> >
> > Paper5567 Authors

---

> > > ### Comment · Reviewer_5yb4 · 2022-11-25
> > > **Follow up reply**
> > >
> > > - I re-edited the review and it should be working now.
> > >
> > > - Regarding Q4, I would suggest first training the samd text+image model twice (denoted the weights as T1+I1 and T2+I2) and then replace T1 with T2 using the proposed method (so the T2 should worknwith I1.) The rational is to separate the contribution of a good image model from a good transferring of the text encoder, and also to have a performance upper bound of the proposed method.

---

> > > > ### Author Response · Authors · 2022-12-02
> > > > **Follow-up Response for Q4**
> > > >
> > > > Dear Reviewer 5yb4,
> > > >
> > > > Thank you so much for your follow-up question. We also agreed that it is a necessary experiment to conduct. So here are our new results on this setting: we have trained two LDM models from scratch (~100 GPU days) with different random seeds based on the filtered Laion dataset (116M). As you suggested, these two models can be denoted as T1+I1 and T2+I2, respectively. Then we attempt to align the T1 and T2 with our proposed MTN models, denoted as T2+MTN+I1 and T1+MTN+I2. This is benchmarked on the COCO-5K following the previous setting. The final results are concluded in the following table:
> > > >
> > > > | Methods| s=5, CLIP |  s=5, FID |  s=7.5, CLIP |  s=7.5, FID |
> > > > | --- | --- | --- |--- | --- |
> > > > |T1+I1|  22.96 | 46.21 | 23.21 | 49.87 |
> > > > |T2+I2|  23.08 | 46.14 | 23.36| 49.66 |
> > > > |T1+I2|  17.88 | 51.32 |18.27  |56.98 |
> > > > |T2+I1|  17.65 | 51.28 | 18.18 | 57.14 |
> > > > |T2+MTN+I1|  22.73 | 47.16 |23.01 | 50.35 |
> > > > |T1+MTN+I2|  22.87 |46.82 | 23.10 | 50.27 |
> > > >
> > > > From the table, we can see that there is a severe performance decrease after replacing the paired text encoder. However, the proposed models (MTN) have greatly healed such damage despite the marginal drop of the paired model, which could be addressed with post-training The new experiment can be strong evidence to support the effectiveness of MTN, and we will include it in our latest manuscript.

---

> > > > > ### Author Response · Authors · 2022-12-06
> > > > > **Follow-up Discussion of Q4**
> > > > >
> > > > > Dear Reviewer 5yb4,
> > > > >
> > > > > Thank you for your valuable suggestions of the comparison between two similar text-to-image models. Please feel free to check our new quantitative results above, and we are happy to discuss further if you still have questions.
> > > > >
> > > > > Regards,
> > > > > Authors of Paper5567

---

### Official Review · Reviewer_jTu3 · 2022-10-25

**Confidence:** 2
**Clarity, Quality, Novelty And Reproducibility:** Lack of novelty, and please revise ot…
**Correctness:** 3
**Technical Novelty And Significance:** 3
**Empirical Novelty And Significance:** Not applicable
**Recommendation:** 6

**Strength And Weaknesses:**

Strength:

1.Good performance.

2.This paper is well written and organized. Motivation is simple and straightforward.

3.As mentioned in the paper, this paper is able to train a model of T2I in an economical way without retraining whole model, and it can even be a multilingual as input as prompt by using XLM-R model.

Weaknesses:
Lack of ablation of text encoder. is it possible to try different text encoder, such as bert, roberta, clip's text encoder, etc.


**Summary Of The Paper:**

This paper proposed an efficient model translation network (MTN) for text-image generation (T2I) by leveraging off-the-shelf pretrained language models with a pre-trained T2I diffusion model, such as T5 and XLM-R. The experiments demonstrate the superiority of this method.

**Summary Of The Review:**

I prefer to accept this paper.

---

> ### Author Response · Authors · 2022-11-19
> **Response to Reviewer jTu3**
>
> Thanks to the reviewer for the recognition of our paper. We sincerely appreciate your valuable comments. Your concern is addressed below.
>
> **Q1: Ablation study of different text encoders.**
>
> Thanks for pointing that out. Yes, our proposed framework is generic to a wide range of text encoders. We have followed your suggestions and tried different language models as the text encoder, including Bert-base (110M #params), Roberta-Large (355M #params), Clip-text (123M #params), T5-large (770M #params) and T5-3B (2.8B #params). All these pre-trained text encoders are aligned with the Latent Diffusion Model (LDM) with our proposed model to replace the original LDM’s text encoder without any finetuning of the LDM UNet.
>
> These plug-and-play models are verified on a randomly selected subset of COCO with 5k samples. We have reported the FID and CLIP scores in the following table for comparison of image quality and image-text alignment. All the model translation networks (MTN) are trained by 18M sentences sampled from the captions of Laion-400M. The testing data are inferred by DDIM with 200 steps. The image size is 256*256. For a comprehensive analysis, we have conducted the experiments on three classifier-free guidance with s=1.5 and 5. The results are summarized as follows.
>
>
> | | CLIP | FID|
> | --- | --- | --- |
> |  Bert-base| 19.42 |37.46  |
> |  Roberta-large| 20.03 |34.06  |
> |  Clip-text| 19.02 | 38.93 |
> |  T5-Large| 19.78 | 35.17 |
> |  T5-3B|**20.67** | **32.80** |
> guidance = 1.5
>
>
>
> | | CLIP  | FID|
> | --- | --- | --- |
> |  Bert-base| 21.97 |55.35  |
> |  Roberta-large| 22.85 | 45.88 |
> |  Clip-text|21.85  | 47.32 |
> |  T5-Large|22.76  |47.91  |
> |  T5-3B|**23.24**  |  **45.48**|
> guidance = 5
>
> From the table, we can notice that both FID and CLIP have increased with the rise of text model sizes which is well aligned with the conclusion of Imagen[1]. The CLIP-text does not perform well here due to the larger domain gap with text-only text models such as Bert. The new manuscript has also been updated to include this in Sec. B.2.
>
> [1] Photorealistic Text-to-Image Diffusion Models with Deep Language Understanding. NeurIPS 22.

---

> > ### Author Response · Authors · 2022-12-02
> > **Follow-up of Discussion**
> >
> > Dear Reviewer jTu3,
> >
> > Thank you so much for your valuable time in reviewing this submission. This is a friendly reminder that the final discussion ends soon. We have tried our best to address your concerns in our responses, which, hopefully, answered your questions. If you have any further concerns, please feel free to let us know.
> >
> > Regards,
> >
> > Authors of Paper5567

---

### Author Response · Authors · 2022-11-19
**Summary of Response**

Thanks to all the reviewers for your time in reviewing this paper. We sincerely appreciate your valuable comments and questions to make this paper better. All the reviewers have recognized the novelty of this paper in both the problem setting and the method. The text-to-image generation receives increasing attention with the development of diffusion models and large-scale training. Efficiency will be one of the most important topics in this area to promote more users and developers.

In general, the reviewers' concerns about this submission can be grouped into three parts:

1) More ablation studies (reviewers jTu3, 5yb4 and 2Z7M). To address it, we have conducted new experiments whose results are summarized in Tab. 3, Tab. 4, and Tab. 5 in Appendix.

2) Comparison with translation models on top of the current diffusion model (reviewers 5yb4 and 2Z7M). To verify the effectiveness of MTN in the multilingual generation, we have introduced the pre-trained translation model (M2M100-418M [1] trained by 7.5B parallel sentences.) for comparison with the results summarized in Tab. 10. Despite the significant deficiency in training data, our proposed MTN achieved promising results compared with the baseline.

3) Analysis of computation cost (reviewers 1TQj,  5yb4, and 2Z7M). Efficiency is one of the major advantages of our proposed method. We have reported the precise computation cost (GPU Days on A100) in Tab. 8 and Tab. 9. For a fair comparison, we have applied T5+LDM trained from scratch as a baseline, which is still inferior to ours despite a lot of training.

Apart from these common questions, we have also detailedly answered the questions of every reviewer below one-by-one. Please let us know if you have additional comments/questions. We will always be happy to discuss.

[1] Beyond English-Centric Multilingual Machine Translation. JMLR 21.

---

### Author Response · Authors · 2022-12-12
**Appreciation Letter**

Dear ACs and Reviewers,

We sincerely appreciate the time and efforts of AC and each reviewer to make this paper better. Your constructive comments and questions during the review process are very helpful for the improvement of paper quality.

We hope our response can address most of the concerns. As the deadline approaches, we are happy to discuss further if you still have any new concerns or questions. Thanks again for your participating!

Regards,
Authors Paper5567

---

### Decision · Program_Chairs · 2023-01-20

**Decision:**

Reject

**Justification For Why Not Higher Score:**

The reviewers appreciated the reviewers for adding additional results to answer the reviewers' concerns. However, several concerns still remain as summarized in the meta review. Thus, on balance, the AC recommends rejection of the paper.

**Justification For Why Not Lower Score:**

N/A

**Metareview: Summary, Strengths And Weaknesses:**

This paper proposes a model translation network (MTN) to address the problem of replacing a text encoder in a text-to-image model without re-training the whole pipeline from scratch. Initially, the paper received scores of 3566. After author rebuttal, the scores have increased to 6666. The AC also held a virtual discussion meeting, where 2 of the reviewers have joined the discussion.

On one hand, this paper is well written, motivation is simple and straightforward, and the authors have made a reasonable effort to show the performance of this method. During rebuttal, the authors provided additional results on comparison to a translation model baseline, analyzed the computational cost, and added additional ablation study.

On the other hand, though all the reviewers gave scores of 6, during the virtual discussion meeting, several concerns still remain. (1) The novelty of the proposed method MTN is limited. It mostly uses standard losses from the domain adaptation literature. (2) It remains unclear how good the MTN works and how much benefit this method brings. For example, a) the image decoder still needs to be fully finetuned in order to have optimal performance. This makes the whole training pipeline complicated, for a somewhat small improvement in FID. b) The human evaluation results in Figure 4 are not convincing enough. It looks like all the 3 methods being tested have roughly the same level of quality, as the percentage difference is minor (though the authors claimed that it can be indeed statistically significant, but may be not significant enough). c) The interesting results of updating a CLIP text encoder of SDM to the multilingual XLM-RoBERTa-L are more contributed to the use of XLM-Roberta-L than the MTN.

On balance, this paper is a borderline case. After virtual discussion meeting,  the AC decides to recommend rejection by the end, and encourages the authors to incorporate all the comments to submit to a future conference.

**Summary Of Ac-Reviewer Meeting:**

The AC held a virtual discussion meeting at Dec. 8th, 9am KST time (i.e., Dec. 7th 6pm Central Time), and 2 of the reviewers attended the meeting. During the discussion, reviewers and the AC had thorough discussion on the pros and cons of the paper. On one hand, the reviewers appreciated the authors to have already added additional results in the paper. On the other hand, the reviewers also commented that it remains unclear how much benefit this method really brings, as summarized in the meta review above. During the virtual discussion meeting, by the end, the AC and the reviewers reached the agreement to recommend rejection.